# HspB8 prevents aberrant phase transitions of FUS by chaperoning its folded RNA-binding domain

Edgar E Boczek[1,2†], Julius Fürsch[3,4†], Marie Laura Niedermeier[3,4], Louise Jawerth[1,5], Marcus Jahnel[1,6], Martine Ruer-Gruß[1], Kai-Michael Kammer[3,4], Peter Heid[3,4], Laura Mediani[7], Jie Wang[1], Xiao Yan[1], Andrej Pozniakovski[1], Ina Poser[1,2], Daniel Mateju[1], Lars Hubatsch[1,5], Serena Carra[7], Simon Alberti[1,6], Anthony A Hyman[1,8], Florian Stengel[3,4*]

[1]Max Planck Institute of Molecular Cell Biology and Genetics, Dresden, Germany; [2]Dewpoint Therapeutics GmbH, Dresden, Germany; [3]University of Konstanz, Department of Biology, Konstanz, Germany; [4]Konstanz Research School Chemical Biology, University of Konstanz, Konstanz, Germany; [5]Max Planck Institute for the Physics of Complex Systems, Dresden, Germany; [6]Biotechnology Center, Technische Universität Dresden, Dresden, Germany; [7]Department of Biomedical, Metabolic and Neural Sciences, University of Modena and Reggio Emilia, Modena, Italy; [8]Center for Systems Biology Dresden (CSBD), Dresden, Germany

**Abstract** Aberrant liquid-to-solid phase transitions of biomolecular condensates have been linked to various neurodegenerative diseases. However, the underlying molecular interactions that drive aging remain enigmatic. Here, we develop quantitative time-resolved crosslinking mass spectrometry to monitor protein interactions and dynamics inside condensates formed by the protein fused in sarcoma (FUS). We identify misfolding of the RNA recognition motif of FUS as a key driver of condensate aging. We demonstrate that the small heat shock protein HspB8 partitions into FUS condensates via its intrinsically disordered domain and prevents condensate hardening via condensate-specific interactions that are mediated by its α-crystallin domain (αCD). These αCD-mediated interactions are altered in a disease-associated mutant of HspB8, which abrogates the ability of HspB8 to prevent condensate hardening. We propose that stabilizing aggregation-prone folded RNA-binding domains inside condensates by molecular chaperones may be a general mechanism to prevent aberrant phase transitions.

**\*For correspondence:** Florian.Stengel@uni-konstanz.de

[†]These authors contributed equally to this work

## Introduction

Condensate formation by liquid-liquid phase separation leads to a local density change of proteins (*Banani et al., 2017*; *Hyman et al., 2014*). Liquid condensates can harden and evolve into less dynamic states with reduced fluidity and protein movement, leading to fibrillar assemblies that are often associated with disease (*Aguzzi and Altmeyer, 2016*; *Patel et al., 2015*; *Molliex et al., 2015*). The molecular changes that underly these aberrant phase transitions, and the ways that cells prevent them, remain poorly understood.

Stress granules have been used as a model to study the role of phase separation in the formation of cellular condensates, as well as disease processes that arise from aberrant phase transitions. Stress granules are composed of RNA and translation factors (*Guillén-Boixet et al., 2020*; *Yang et al., 2020*; *Sanders et al., 2020*; *Jain et al., 2016*; *Markmiller et al., 2018*; *Youn et al., 2018*). There is an increasing body of evidence that the ability of stress granule proteins to phase

separate forms the basis for stress granule assembly (*Guillén-Boixet et al., 2020*; *Yang et al., 2020*; *Sanders et al., 2020*). For instance, purified stress granule residing RNA-binding proteins (RBPs) phase separate into liquid droplets in vitro. Reconstituted droplets have physicochemical properties similar to stress granules in cells (*Patel et al., 2015*; *Guillén-Boixet et al., 2020*; *Yang et al., 2020*; *Sanders et al., 2020*). In vitro reconstituted molecular condensates are metastable and age into less dynamic amorphous drops, and fibrillar aggregates with time (*Patel et al., 2015*). These aggregates are reminiscent of protein aggregates seen in patients afflicted with age-related diseases such as amyotrophic lateral sclerosis (ALS) and frontotemporal dementia (FTD). This process is referred to as molecular aging (*Patel et al., 2015*). Notably, the molecular aging process is accelerated by ALS-linked mutations in FUS and other RBPs (*Patel et al., 2015*; *Molliex et al., 2015*). This suggests that molecular aging of liquid condensates such as stress granules can be a disease process.

Recent studies have provided evidence for a link between stress granules and small heat shock proteins (sHSPs) (*Mateju et al., 2017*; *Ganassi et al., 2016*; *Liu et al., 2020*; *Yu et al., 2021*; *Gu et al., 2020*). These ATP-independent chaperones hold unfolded proteins in a refolding-competent state and both their intrinsically disordered region (IDR) and folded α-crystallin domain (αCD) contribute to this activity (*Haslbeck et al., 2019*). sHSPs have been shown to accumulate in stress granules that undergo an aberrant conversion from a liquid to a solid-like state (*Mateju et al., 2017*; *Ganassi et al., 2016*). Aberrant stress granules have also been linked to disease and many sHSPs are associated with neurodegenerative disorders (*Vendredy et al., 2020*). This suggests that chaperones, such as sHSPs, may regulate the properties of stress granules and presumably also the molecular aging process of stress granule proteins such as FUS.

FUS has been a model protein to study both aberrant and physiological phase transitions (*Patel et al., 2015*; *Guo et al., 2018*; *Hofweber et al., 2018*; *Qamar et al., 2018*; *Wang et al., 2018*; *Yoshizawa et al., 2018*; *Kato et al., 2012*). It contains an intrinsically disordered prion-like low complexity domain (LCD) composed of only a small subset of amino acids, an RNA-binding domain (RBD) containing intrinsically disordered RGG-rich motifs, a Zinc finger (ZnF), and a folded RNA recognition motif (RRM). It has been shown that phase separation of FUS family proteins is driven by multivalent interactions among tyrosine residues within its LCD and arginine residues of their RBDs (*Qamar et al., 2018*; *Wang et al., 2018*; *Schuster et al., 2020*). At least part of the LCD of FUS can assemble into amyloid fibrils (*Murray et al., 2017*) and the isolated FUS-LCD can adopt a hydrogel-like state that also depends on amyloid-like interactions (*Kato et al., 2012*). By contrast, the LCD appears to be disordered in liquid droplets, exhibiting no detectable secondary structure (*Murthy et al., 2019*). Similar observations have been made for the LCD of hnRNPA2 (*Xiang et al., 2015*). In addition to the LCD, the isolated FUS-RRM has been shown to spontaneously self-assemble into amyloid fibrils (*Lu et al., 2017*). However, the molecular mechanism by which proteins such as FUS undergo molecular aging are still unknown.

To determine the molecular changes during the aging of stress granule proteins such as FUS, it is critical to monitor protein-protein interactions (PPIs) and conformational dynamics within condensates. However, this has remained a major challenge. We and others showed previously that chemical crosslinking coupled to mass spectrometry (XL-MS) is well suited to map PPIs and that relative changes in crosslinking as probed by quantitative XL-MS (qXL-MS) can provide a structural understanding of protein dynamics (*Yu and Huang, 2018*; *Sailer et al., 2018*; *Walker-Gray et al., 2017*; *Patel et al., 2017*). Here, we adopt XL-MS to study condensates.

In this study, we take a biochemical approach using purified proteins to reconstitute a chaperone-mediated quality control mechanism associated with RNP granules and combine it with qXL-MS to probe PPIs inside condensates. We find that unfolding of the RRM drives FUS aging and that interaction of its folded RRM with the sHSP HspB8 slows down this aging process. Importantly, these condensate-specific interactions are altered in a disease-associated HspB8 mutant, resulting in its inability to prevent FUS aging.

## Results

### Quantitative and time-resolved XL-MS reveal domain-specific changes in crosslink abundances underlying condensate formation

To investigate the condensate-specific interactions of FUS after phase separation, we diluted purified FUS-G156E protein (*Patel et al., 2015*; *Nomura et al., 2014*), called FUS$_m$ throughout this manuscript, either into a low salt solution, which induces phase separation or into a high salt buffer, which prevents phase separation. After crosslinking of lysine residues and subsequent digestion, equal amounts of peptides were subjected to mass spectrometry analysis to reveal condensate-specific crosslink patterns (*Figure 1A*, *Figure 1—figure supplement 1A*, *Supplementary file 1*). These crosslinks can either reflect the spatial proximity of regions and protein-domains within a given protein, called intra-links, or between different proteins, called inter-links. For a detailed description of the different crosslink types *see Figure 1—figure supplement 1B*. When the crosslinker reacts twice within one peptide, this is called a loop-link. Additional information comes from crosslinking of one side of the crosslinker with the protein and hydrolysis on the other side. This is called a mono-link and reveals information on the accessibility of a specific lysine residue. In some cases, we can also follow links between proteins of the same species (homo-dimeric link). These are crosslinks between overlapping peptides whose sequence is unique within the protein and that must therefore originate from different copies of the same protein.

Please note that the crosslinker used throughout this manuscript, disuccinimidyl suberate (DSS), will react with primary amines and thus primarily link lysine residues within studied proteins. It is important to remember that the LCD of FUS contains no lysines and therefore our approach is not picking up interactions between and amongst the LCD domains (for information on the LCD of FUS see lysine-rich variant FUS_K9 in *Figure 2—figure supplement 1G*).

The most prominent feature we detected after FUS$_m$ condensation was that multiple intra-links within the RRM domain of FUS$_m$ increased inside the condensates. This suggests that there are augmented contacts within the RRM domain, indicative of a structural change. In addition, there was an increase in homo-dimeric links between RRM domains, indicative of interactions between RRM domains of different FUS$_m$ molecules. Additionally, links between the RRM domain and the nuclear localization sequence (NLS) were increased, while links within the Zinc-finger domain were decreased (*Figure 1B*). Taken together, these results suggest structural changes in the RRM domain accompany condensation.

### Crosslinks within the RRM domain change during molecular aging

It has been shown previously that FUS$_m$ (FUS-G156E) shows an increased propensity for aggregation in vitro and in vivo (*Nomura et al., 2014*), resulting in slowed down internal dynamics over time (*Patel et al., 2015*; *Nomura et al., 2014*; *Jawerth et al., 2020*; *Jawerth et al., 2018*) and enabling us to study molecular aging of FUS condensates in vitro. To study protein dynamics and conformational changes during aging, we devised a time-resolved, quantitative XL-MS approach (*Figure 1C*) based on significantly shortened crosslinking times than conventionally used (*Iacobucci et al., 2019*; *Figure 1—figure supplement 1C*), which put us into the position to assess all stages of the aging process (*Figure 1—figure supplement 1D*). We then monitored the conversion of fresh FUS$_m$ condensates into fibers by fluorescence microscopy (*Figure 1C*). At indicated time points, the assemblies were crosslinked and analyzed by MS. We looked at 11 time points during the aging process and consistently quantified 77 crosslinks relative to fresh condensates (T1) (*Supplementary file 2*). A global view of all quantified crosslinking sites shows that the vast majority of changes are happening during the formation of fiber states (*Figure 1—figure supplement 1E*). A closer examination focusing on those crosslinks within the RRM that were increased during condensation shows that these also change during molecular aging and particularly during fiber formation (*Figure 1C* and *Figure 1—figure supplement 1F*). For instance, the link within the RRM bridging positions 339 and 362 decreases as fibers form. The general trend is for interactions between RRM domains and within RRM domains to decrease as condensates age.

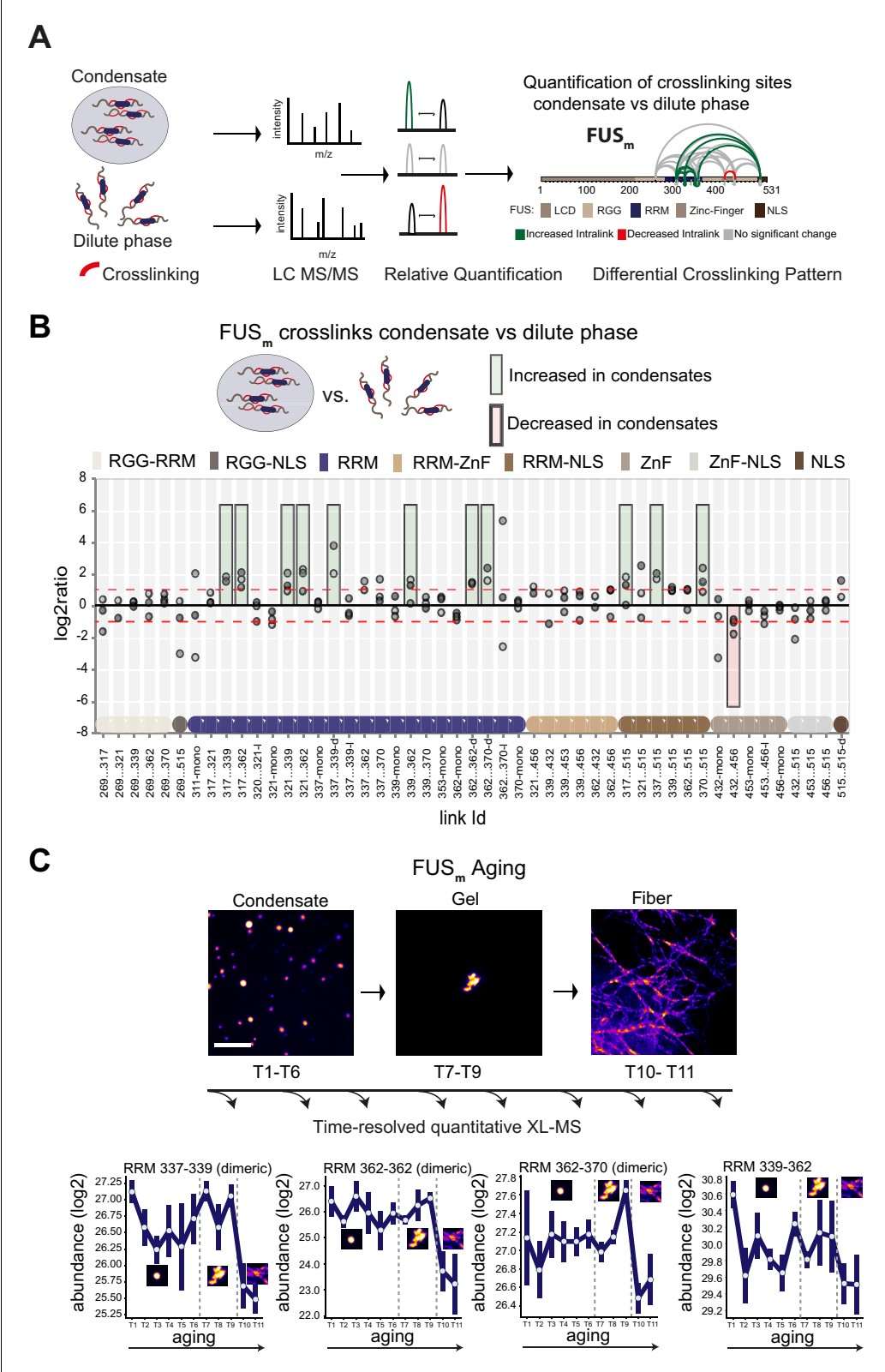

**Figure 1.** Domain-specific changes in crosslink abundances underlying condensate formation and molecular aging. (**A**) Workflow of quantitative crosslinking coupled to mass spectrometry (qXL-MS) of FUS$_m$ condensates. (**B**) Crosslink abundance plot from reconstituted FUS$_m$ condensates. Plotted are the relative enrichment (droplet vs. non-droplet) for each unique crosslinking site (y-axis) sorted according to the known domain structure within FUS$_m$ (x-axis). Shown are only high confidence crosslinking sites (see Materials and methods for details) from three biologically independent sets of

*Figure 1 continued*

experiments (n=3; circles in different shades of gray). Please note that FUS$_m$ used throughout this manuscript contains both a C-terminal GFP used for visualization and a 5 AA N-terminal tag used for purification. All uxIDs therefore have an offset of 5 AA compared to the UniProt entry for human FUS (P35637). Crosslinking sites that were consistently upregulated or downregulated twofold or more (log2ratio$\geq$1 or $\leq-1$ and FDR$\leq$0.05) in at least two out of three biological replicate sets and in addition contained no opposing regulation in any replicate set were considered significant and are highlighted with a green (enriched in droplets) or red background rectangle (decreased in droplets). All other changes in crosslinking abundances were considered insignificant and are shown on gray background. The significance threshold of twofold enrichment is indicated as dashed red line. Dimeric links are indicated by an additional '−d', loop-links by a '−l' and mono-links by an '-mono' at their respective unique crosslinking site. Domain structures within FUS$_m$ are color-coded as in (**A**). RGG refers to AAs 220–289. (**C**) *Upper panel*: workflow time-resolved quantitative XL-MS. The conversion of fresh FUS$_m$ condensates via the gel state into fibers was monitored by fluorescence microscopy. Scale bar is 10 µm. At indicated time points, aliquots of the stock solution were crosslinked for 5 min, flash-frozen in liquid nitrogen, and subsequently analyzed by MS (see Materials and methods for details). *Lower panel*: shown are changes of RRM crosslinks during aging that were increased during condensation (**B**). The logarithmic total MS1 area for each time point during aging is plotted (SDs; n=6). Domain structures within FUS$_m$ are color-coded as in (**A**). RRM, RNA recognition motif.

The online version of this article includes the following figure supplement(s) for figure 1:

**Figure supplement 1.** Quantitative and time-resolved XL-MS reveal domain-specific changes in crosslink abundances underlying condensate formation.

## The small heat shock protein HspB8 partitions into FUS condensates and interacts with the RRM domain

To test whether these changes in links are indeed driven by aging, we next looked for ways to slow down the aging process. sHSPs are ATP-independent chaperones and are structurally divided into IDRs and a folded chaperone domain called αCD (*Carra et al., 2019*). HspB8 has previously been localized to stress granules (*Ganassi et al., 2016*). We therefore purified HspB (*Figure 2—figure supplement 1A*) and looked at its interaction with FUS$_m$ condensates in vitro. We find that HspB8 was sequestered into reconstituted FUS$_m$ condensates (*Figure 2A*) and that its fluorescence signal superimposed with the FUS$_m$-GFP signal (*Figure 2B*), while it did not form droplets on its own under these conditions (*Figure 2—figure supplement 1B*). Another closely related sHSP that has been localized to stress granules (*Mateju et al., 2017*), HspB1, did not accumulate inside FUS$_m$ droplets (*Figure 2C*, *Figure 2D*, *Figure 2—figure supplement 1A*). Using the fluorescence signal of labeled HspB8 and a calibration curve, we determined the concentration of HspB8 inside FUS$_m$ condensates to be 2.4 mM (*Figure 2—figure supplement 1C and D*).

We then used XL-MS, to probe PPIs between FUS$_m$ and HspB8 inside the condensates. Inter-links (crosslinks between different proteins) predominantly formed inside the dense phase of the condensates (*Figure 2E*, *Supplementary file 3*). Together with data from thermophoresis binding experiments (*Figure 2—figure supplement 1E*), this suggests that HspB8 specifically binds to FUS$_m$ inside and not outside condensates. We find the majority of inter-links within the droplets were formed between the αCD of HspB8 and the RRM of FUS$_m$, and to a lesser degree the FUS-NLS. This condensate-specific and highly reproducible crosslinking pattern (*Figure 2E*, *Figure 2—figure supplement 1F and G*) was not seen when lactalbumin, a FUS-unrelated molten globule protein that unspecifically partitions into FUS$_m$ condensates (*Figure 2—figure supplement 1H and I*) was used as a control. Taken together, interactions between HspB8 and FUS$_m$ show a condensate-specific increase; in particular, interactions between the αCD of HspB8 and the FUS$_m$-RRM are significantly upregulated inside condensates (*Figure 2F*).

## HspB8 prevents hardening and fiber formation of FUS droplets and keeps them dynamic

A time-course experiment using photobleaching recovery (fluorescence recovery after photobleaching [FRAP]) revealed that in the presence of HspB8, FUS$_m$ retained its liquidity over a time span of 24 hr (*Figure 3A–C*, *Figure 3—figure supplement 1A–D*). FUS$_m$ droplets were able to fuse even after 6 hr (*Figure 3D*): the relaxation times of the fusion events were not affected (*Figure 3E*) and the drops no longer adhered to each other (*Figure 3—figure supplement 1E*). In addition to protecting fresh FUS$_m$ condensates from converting into fibers (*Figure 3F*), the chaperone prevented further conversion of pre-aged FUS$_m$ droplets, and prevented seeding of fiber growth (*Figure 3G*, *Video 1*). This correlated with the localization of HspB8 to FUS$_m$ assemblies

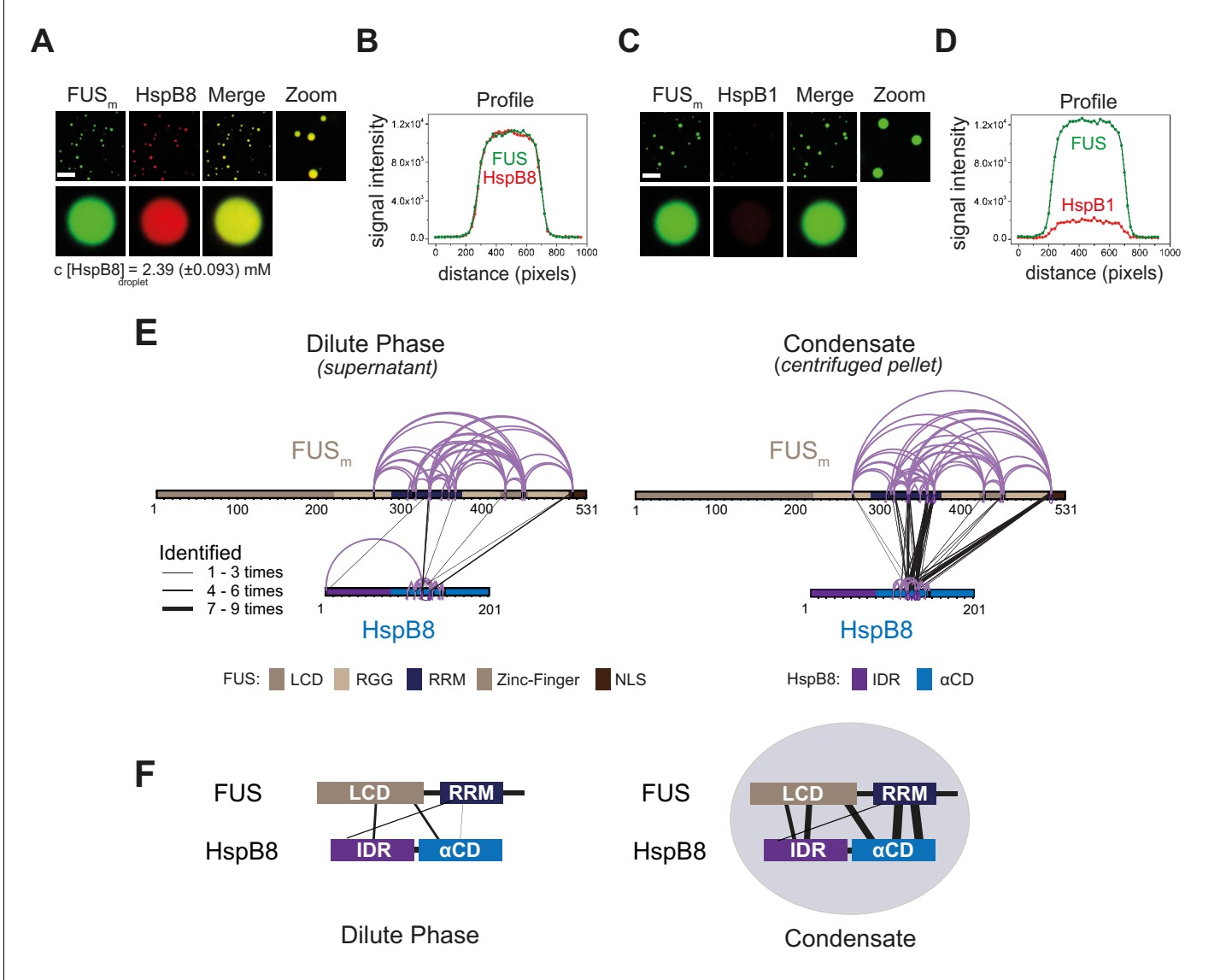

**Figure 2.** The small heat shock protein HspB8 partitions into FUS condensates and interacts with the RRM domain. (A) 0.25 μM Cy3-labeled HspB8, 4.75 μM unlabeled HspB8 (1:20 mix), and 5 μM FUS$_m$-GFP were mixed in low salt buffer and the resulting droplets were imaged in a confocal microscope. Scale bar is 10 μm. (B) Fluorescence intensity plot profile spanning a FUS$_m$-HspB8 droplet in (A). (C) 0.25 μM Cy3-labeled HspB1, 4.75 μM unlabeled HspB1 (1:20 mix), and 5 μM FUS$_m$-GFP were mixed in low salt buffer and imaged in a confocal microscope. Scale bar is 10 μm. (D) Fluorescence intensity plot profile spanning a FUS$_m$-HspB1 droplet in (C). (E) Overall crosslinking pattern of mixtures of FUS$_m$ and HspB8 that were crosslinked under condensate-inducing low salt conditions (75 mM) and separated into the dilute phase (*left*) and dense phase of the condensates (*right*) by centrifugation. Experiments were carried out in three biologically independent sets of experiments (meaning separate batches of expressed protein). For one set of experiments, each sample was independently crosslinked in triplicates and crosslinks were only considered, if they were identified in two out of three replicates with a deltaS<0.95, a minimum Id score≥20, and an Id score≥25 in at least one replicate (filtering was done on the level of the unique crosslinking site) and an FDR≤0.05. Inter-links are shown in black and the total number of identifications is indicated by the thickness of the connection. Intra-links are shown in violet, mono-links with a flag, loop links with a pointed triangle, and homo-dimeric links with a loop. (F) Representative overview of observed crosslinks between the LCD and RRM domains in FUS$_m$ and the IDR and αCD domains in HspB8 in the dilute and the dense phase as detected by XL-MS (based on (E), *Figure 2—figure supplement 1F and G*). αCD, α-crystallin domain; IDR, intrinsically disordered region; LCD, low complexity domain; RRM, RNA recognition motif.

The online version of this article includes the following source data and figure supplement(s) for figure 2:

**Figure supplement 1.** The small heat shock protein HspB8 partitions into FUS condensates and interacts with the RRM domain.

**Figure supplement 1—source data 1.** Uncropped image for *Figure 2—figure supplement 1A*.

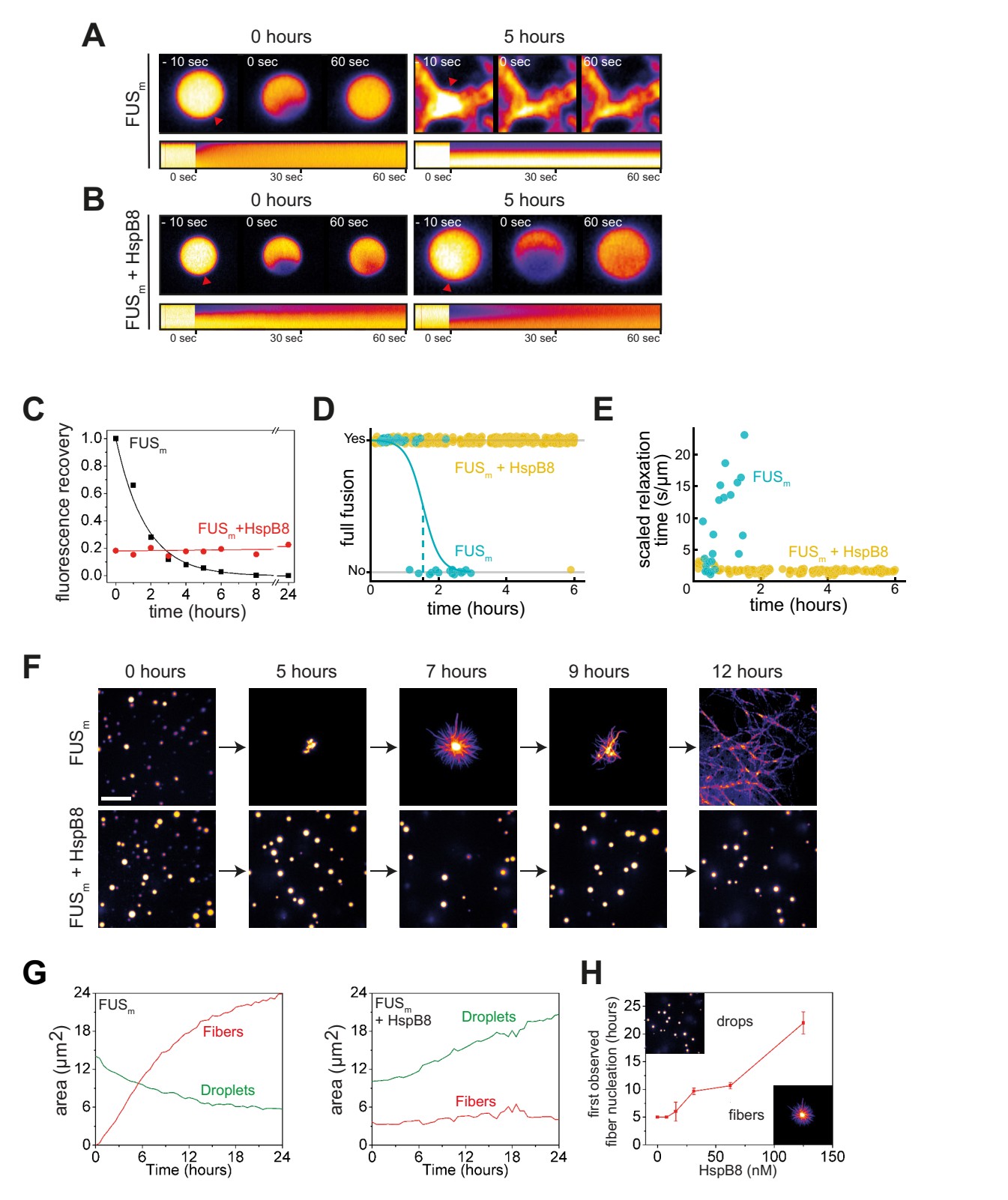

**Figure 3.** HspB8 prevents hardening and fiber formation of FUS droplets and keeps them dynamic. (**A**) FRAP experiment of fresh $FUS_m$ condensates (0 hr) and condensates incubated for 5 hr. A kymograph shown below illustrates the kinetics of the process. (**B**) FRAP experiment of fresh $FUS_m$ condensates mixed with HspB8 (0 hr) and condensates incubated for 5 hr. A kymograph shown below illustrates the kinetics of the process. (**C**) Kinetics of the $FUS_m$ aging process. Plotted are the initial slopes of the FRAP recovery curves for $FUS_m$ condensates in the absence (black) or presence of HspB8

*Figure 3 continued on next page*

*Figure 3 continued*

(red). (D) Successful complete fusion events were registered over time, demonstrating the aging process of the FUS$_m$ sample in the absence (turquoise, N=40) or the presence of HspB8 (yellow, N=330). The half-life of liquid-like FUS$_m$ condensates alone was estimated to be around 1.5 hr from logistic regression. (E) The size-normalized coalescence relaxation time is an indicator for the material state of the condensates. While it increases for FUS$_m$ condensates during the hardening process, it stays constant over 6 hr in the presence of HspB8. Tweezer experiments were performed with fresh samples of 5 µM FUS$_m$ with or without 20 µM HspB8. (F) Aging process of 5 µM FUS$_m$ condensates in the absence and presence of 5 µM HspB8. In the presence of the chaperone, the droplet morphology is maintained over the whole timeframe of the experiment (12 hr). Scale bar is 10 µm. (G) *Left panel:* shown is the total area of droplet material (green line) or fibrous material (red line) within FUS$_m$ droplets as a function of time after FUS$_m$ droplets were added to the existing fibrous material. Spacing between data points is 30 min. *Right panel:* shown is the total area of droplet material (green line) and fibrous material (red line) within FUS$_m$ droplets as a function of time after FUS$_m$ droplets and HspB8 were added to the existing fibrous material. Spacing between data points is 30 min. (H) The onset of 5 µM FUS$_m$ fiber formation as a function of HspB8 concentration. FRAP, fluorescence recovery after photobleaching.

The online version of this article includes the following figure supplement(s) for figure 3:

**Figure supplement 1.** HspB8 prevents hardening and fiber formation of FUS droplets and keeps them dynamic.

(*Figure 3—figure supplement 1F*). The protective effect of HspB8 on FUS$_m$ was also observed at sub-stoichiometric HspB8 concentrations (*Figure 3H*).

In order to investigate the molecular effect of HspB8 on the hardening of FUS$_m$, we conducted quantitative XL-MS experiments in the presence of HspB8 and looked at crosslinks between FUS$_m$ peptides. We found that previously observed crosslink patterns for pure FUS$_m$ droplets, in particular upregulated crosslinks of the FUS-RRM, were still seen in the presence of HspB8 (*Figure 3—figure supplement 1G*, *Supplementary file 4*). However, interactions between both the RRM with RGG and the RRM with the ZnF decreased, as did links within the ZnF and between the ZnF and the NLS. Our data show that HspB8 binds to the FUS-RRM also during prolonged incubation times via the αCD::RRM interface established during condensation (*Figure 3—figure supplement 1H & I*). This suggests that HspB8 binds to the RRM domain in condensates and prevents it from forming aberrant interactions with other domains in the protein.

## The disordered region of HspB8 directs the αCD into FUS condensates for chaperoning

sHSPs consist of an αCD and flanking regions that are thought to be intrinsically disordered (*Sudnitsyna et al., 2012*). Disorder prediction and circular dichroism analysis revealed that HspB8 is likely to be significantly more disordered than the closely related HspB1 (*Figure 4—figure supplement 1A–D*). To dissect the influence of the conserved αCD and the disordered IDR on the aging process of FUS$_m$ we designed SNAP fusion constructs where we fused either only the IDR, the αCD, or full-length HspB8 to a fluorescently labeled SNAP-tag (*Figure 4A*). While the IDR-SNAP fusion construct was still recruited to FUS$_m$ condensates, the αCD-SNAP variant no longer partitioned (*Figure 4B*). A decreased partitioning of the IDR-SNAP compared to the full-length construct indicated a contribution of the αCD (*Figure 4B*, *Figure 4—figure supplement 1E*). The IDR-SNAP construct was also not active in preventing FUS$_m$ fiber formation (*Figure 4C*), while the αCD-SNAP construct showed slight activity at high concentrations (*Figure 4—figure supplement 1F*). Similarly, in a FRAP assay, the αCD-SNAP construct showed slight activity, while IDR-SNAP

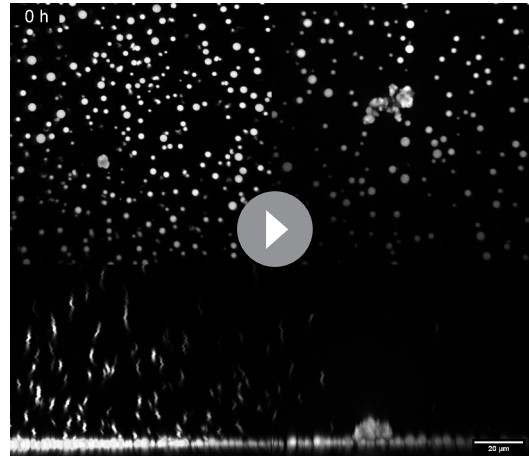

**Video 1.** Time lapse of FUS fiber growth. Time-lapse movie of FUS$_m$ fiber growth in the presence and absence of HspB8. The y-direction images are scaled by a factor of 3 to match the scale bar for the other maximum projection. The histograms/contrast are automatically set using the enhance contrast command in FIJI with 0.3% saturated pixels.
https://elifesciences.org/articles/69377#video1

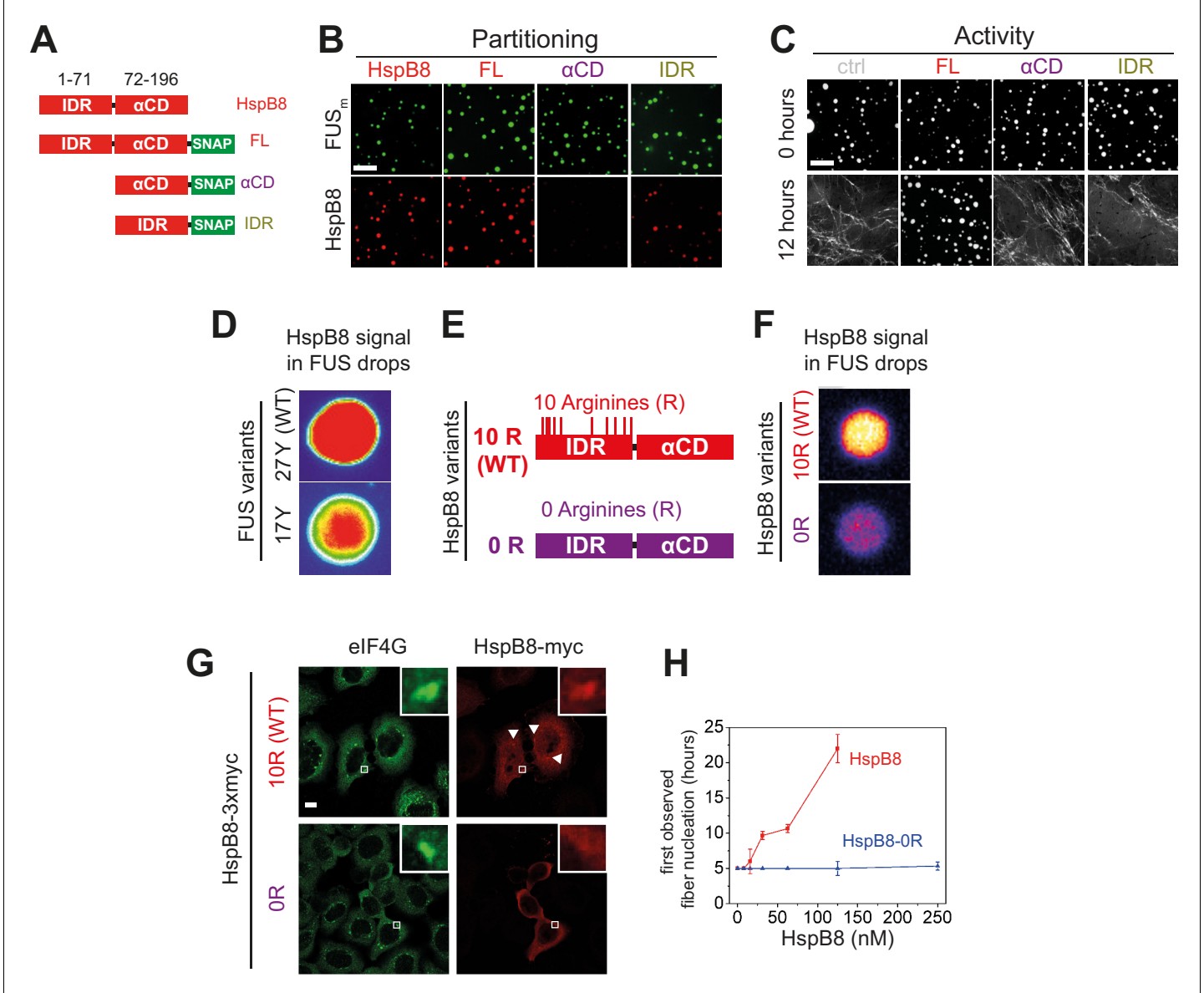

**Figure 4.** Arginines within the disordered region of HspB8 direct the α-crystallin domain into FUS condensates for chaperoning. (**A**) Overview of HspB8 truncation variants used in this study. FL (full-length HspB8 fused to a SNAP-tag), αCD (HspB8 alpha-crystallin domain [AA72–196] fused to a SNAP-tag), and IDR (HspB8-N-terminus [AA1–71] fused to a SNAP-tag). (**B**) Partitioning of 5 μM HspB8 SNAP constructs (1:20 mix of labeled:unlabeled) into 5 μM FUS$_m$ condensates. Scale bar is 10 μm. (**C**) Aging assay of 5 μM FUS$_m$ condensates in the absence (ctrl) and presence of 5 μM HspB8 truncation variants. Scale bar is 10 μm. (**D**) Partitioning of 5 μM HspB8 (1:20 mix of labeled:unlabeled) into condensates formed by 5 μM FUS wildtype (27 Y, WT) or a variant with a reduced number of tyrosines in its IDR (17 Y). (**E**) Location of Arg residues in the primary structure of HspB8. 10 Arg residues are located in the N-terminus of HspB8-WT (10 R). In the HspB8-0R variant, these Arg residues are replaced by Gly residues (0 R). (**F**) Partitioning of 5 μM HspB8-WT or HspB8-0R variant (1:20 mix of labeled:unlabeled) into condensates formed by 5 μM FUS$_m$. (**G**) HeLa Kyoto cells expressing HSPB8-WT-3xmyc or HSPB8-0R-3xmyc from a plasmid were subjected to heat shock at 43.5°C for 1 hr. Cells were fixed and stained with myc and eIF4G specific antibodies. Merged image composed of eIF4G (green) and myc (red) signals is shown. (**H**) The onset of 5 μM FUS$_m$ fiber formation as a function of HspB8-0R concentration. IDR, intrinsically disordered region.

The online version of this article includes the following figure supplement(s) for figure 4:

**Figure supplement 1.** Arginines in the disordered region of HspB8 direct the αCD into FUS condensates.

was inactive in preventing FUS$_m$ gelation (*Figure 4—figure supplement 1G*). A partitioning analysis of these variants with droplets formed only by the LCD of FUS (AA1–211) under crowding conditions revealed a similar partitioning pattern as compared to the full-length FUS$_m$ (*Figure 4—figure supplement 1H*), indicating that the HspB8-IDR interacts with the FUS-LCD. In order to rule out potential perturbations of HspB8 domain activity by fusion to the SNAP moiety, we designed additional swap variants of HspB8 and the closely related HspB1 (*Figure 4—figure supplement 1I*) and all our experiments with the HspB8-HspB1 swap variants mirrored the results with the HspB8-SNAP fusions (*Figure 4—figure supplement 1J-N*).

In summary, these results suggest that the IDR of HspB8 targets the αCD to condensates via interaction with the LCD of phase separated FUS and that the HspB8-αCD is the active domain in preventing FUS aging.

## Arginines in the disordered region of HspB8 direct the αCD into FUS condensates

Recent studies identified interactions between tyrosines in the LCD and arginines in the RBD of FUS to be crucial for its ability to phase separate (*Qamar et al., 2018*; *Wang et al., 2018*; *Schuster et al., 2020*). Because HspB8 partitions into condensates formed only by the FUS-LCD (*Figure 4—figure supplement 1N*), we hypothesized that HspB8 could interact with tyrosines in the LCD of FUS. We tested the partitioning of HspB8 into a variant of FUS with a decreased number of tyrosines in the FUS-LCD. FUS wildtype (WT) has 27 tyrosines in its LCD. Mutating all of these to serines in the 0 Y variant abrogates phase separation, but the 17 Y FUS variant still undergoes phase separation (*Wang et al., 2018*; *Figure 4D*). Partitioning of HspB8 was significantly reduced for FUS condensates formed by the 17 Y variant (*Figure 4D*), suggesting that tyrosines in the FUS-LCD interact with HspB8. HspB8 has 10 arginines in its predicted IDR (*Figure 4E*). We exchanged these for glycines resulting in the 0 R variant of HspB8. This variant did not accumulate in FUS$_m$ droplets in vitro (*Figure 4F*) and did not partition into stress granules in cells, contrary to the behavior of HspB8-WT (*Figure 4G*). When we tested the 0 R variant of HspB8 in a FUS$_m$ aging assay we found that its ability to prevent fiber formation was completely abolished (*Figure 4H*).

## RRM unfolding drives FUS aging and is rescued by HspB8

Next, we sought to test the consequences of the interaction between the HspB8-αCD and the FUS-RRM for FUS aging. To this end, we deleted the RRM in the variant FUS$_m$ΔRRM (ΔAA285–371) and monitored its aging process in the presence and absence of HspB8 (*Figure 5A*, *Figure 5—figure supplement 1A*). First, FUS$_m$ aging was observed after 4 hr in the control condition, while co-incubation with HspB8 prevented the aging process over the time course of the experiment. Deletion of the RRM domain in FUS$_m$ΔRRM significantly slowed down aging and first aggregates were observed after 36 hr. Remarkably, HspB8 was not able to prevent aging of the FUS$_m$ΔRRM variant, indicating that HspB8 binding to the FUS-RRM is required for HspB8 to act as a chaperone for FUS. Our XL-MS data shows that the majority of inter-links between FUS$_m$ and HspB8 that were significantly increased within condensates in fact formed between the αCD of HspB8 and the RRM of FUS$_m$ (*Figure 5B*). Thus, while the RRM domain does not seem to be solely responsible for FUS aging, it significantly contributes to the initiation of the process.

The RRM domain of FUS is a folded domain in an otherwise disordered protein (*Figure 5—figure supplement 1B*). We suspected that RRM unfolding might serve as a seed for the formation of FUS aberrant conformations that would initiate FUS aggregation and fiber formation. To test this hypothesis, we performed a heat shock experiment to unfold the RRM domain of FUS (*Figure 5C*). The melting temperature of the isolated FUS-RRM has been reported to be 52℃ (*Lu et al., 2017*). We prepared reactions of condensates formed by full-length FUS or FUS$_m$ΔRRM and incubated these for 10 min at 55℃ to unfold the RRM. At this temperature, FUS$_m$ condensates were dissolved (*Figure 5C*). We then cooled down the reactions to 25℃ and assessed the reactions by fluorescence microscopy. While full-length FUS$_m$ formed amorphous aggregates after cooling down from heat shock, the FUS$_m$ΔRRM variant condensed into spherical droplets (*Figure 5C*). This result strongly suggests that unfolding of the RRM domain is an integral part of the aging process and deletion of the RRM prevents temperature-induced aggregation of FUS$_m$ condensates.

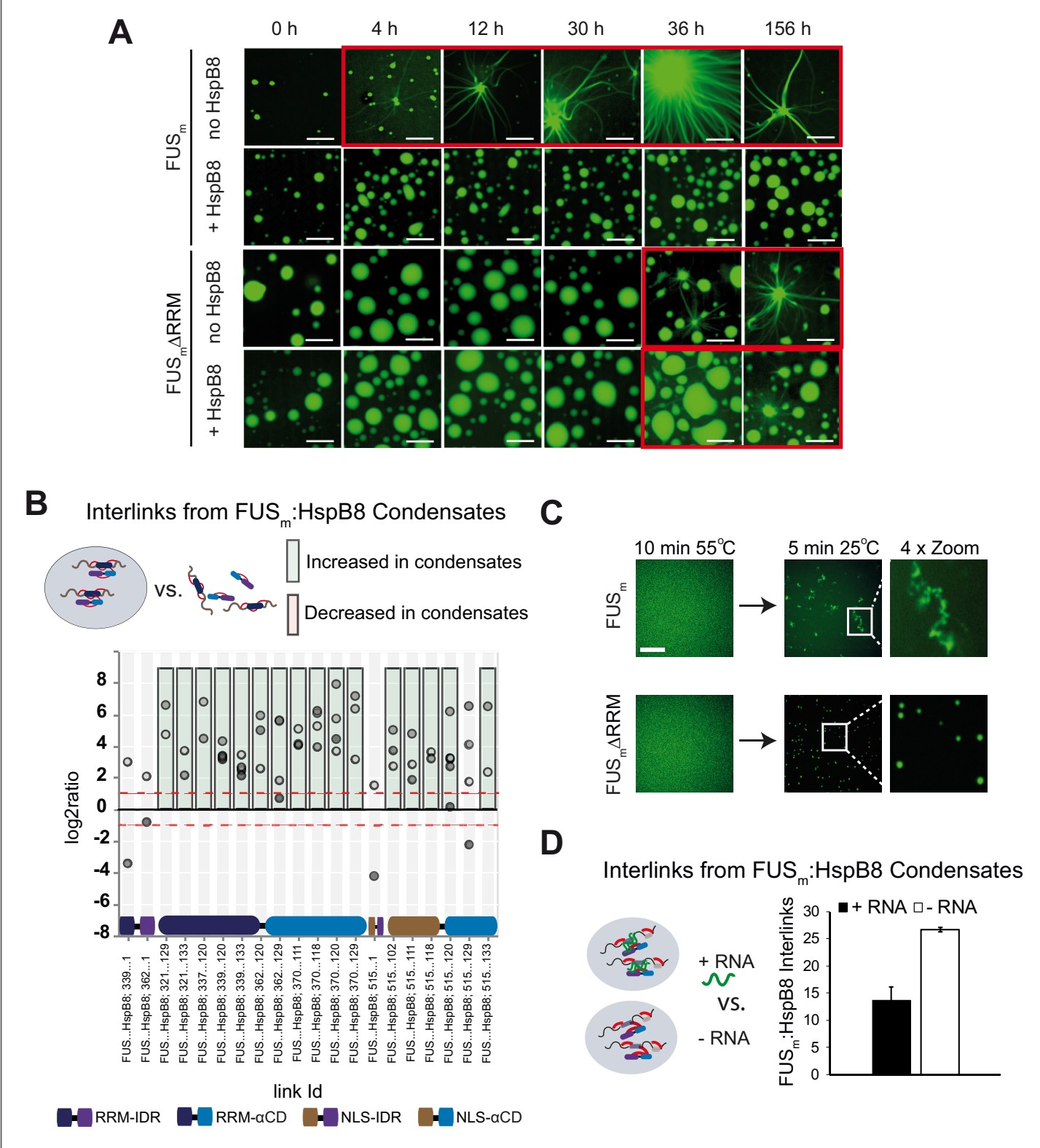

**Figure 5.** RRM unfolding drives FUS aggregation and is rescued by HspB8. (**A**) Aging process of molecular condensates containing either FUS$_m$ or the FUS$_m$ΔRRM (ΔAA285–371) variant in the presence or absence of HspB8 monitored by fluorescence microscopy over time. Scale bar is 10 μm. (**B**) Interlink abundances from reconstituted FUS$_m$:HspB condensates. Plotted are the relative enrichment (droplet vs. non-droplet) for each unique crosslinking site (y-axis) sorted according to the known domain structure within FUS$_m$ and HspB8 (x-axis). Shown are only high confidence crosslinking sites (see Materials and methods for details) from five biologically independent sets of experiments (n=5; circles in different shades of gray). Crosslinking sites

*Figure 5 continued on next page*

*Figure 5 continued*

that were consistently upregulated or downregulated twofold or more (log2ratio≥1 or ≤−1 and FDR≤0.05) in at least two out of five biological replicate sets of experiments and in addition contained no opposing regulation in any replicate set were considered significant and are highlighted with a green (enriched in droplets) or red background rectangle (decreased in droplets). All other changes in crosslinking abundances were considered insignificant and are shown on gray background. The significance threshold of twofold enrichment is indicated as dashed red line. (C) Fluorescence microscopy images of FUS_m and FUS_mΔRRM variant during and after incubation under heat shock conditions. Samples were heated to 55˚C for 10 min followed by a 5 min cool-down step to 25˚C. Scale bar is 10 μm. (D) FUS_m:HspB8 condensates were crosslinked in the presence (black) or absence (white) of a customized RNA oligonucleotide previously shown to bind to FUS (*Maharana et al., 2018*) and analyzed by LC-MS/MS (n=3; FDR≤0.05). RRM, RNA recognition motif.

The online version of this article includes the following figure supplement(s) for figure 5:

**Figure supplement 1.** RRM unfolding drives FUS aging and is rescued by HspB8.

It has been shown that RNA can bind to the FUS-RRM, prevent aging of FUS and dissolve condensates at high concentrations (*Kramer et al., 2014*; *Maharana et al., 2018*). Hence, we were wondering whether RNA exerts these effects by virtue of binding and stabilizing the RRM domain. We tested this by looking at competition between RNA and HspB8 binding to FUS_m using XL-MS and crosslinked condensates of FUS_m and HspB8 in the presence or absence of RNA. While the addition of RNA led to a significant decrease in the number of detected inter-links between FUS_m and HspB8, it did not alter the number of intra-links within FUS_m or HspB8 (*Figure 5D* and *Figure 5—figure supplement 1C*, *Supplementary file 5*). This result suggests that RNA and HspB8 can compete for binding to FUS_m and indicates a similar mechanism by which they stabilize its RRM domain to prevent FUS_m aging.

## A disease-associated mutation interferes with HspB8 activity

Mutations of the lysine 141 residue in the αCD of HspB8 have been associated with Charcot-Marie Tooth disease, a currently incurable dominant autosomal disorder of the peripheral nervous system leading to muscular dystrophies (*Irobi et al., 2004*; *Figure 6A*). The mechanistic cause of the disease is still enigmatic, although experimental evidence indicates a decreased chaperone activity for HspB8-K141E (*Kim et al., 2006*; *Carra et al., 2010*; *Carra et al., 2008*). We introduced the K141E mutation into HspB8 and when we tested HspB8-K141E for localization to FUS_m droplets, we found that it still partitioned into reconstituted FUS_m droplets (*Figure 6B*). While the HspB8 WT was active in a FUS_m aging experiment, HspB8-K141E could not prevent FUS_m aging (*Figure 6C*) and when mixed with the WT, the mutant exerted a dominant-negative effect over the WT, preventing the WT from being active. By using FRAP, we found that the mutant was much less effective compared to the WT in keeping FUS_m in a dynamic state (*Figure 6D*). Remarkably, the WT-mutant mix showed even lower chaperone activity, underlining the dominant-negative role of HspB8-K141E mutant over the WT (*Figure 6D*).

We then performed XL-MS using the HspB8-K141E mutant. In comparison to HspB8-WT, the mutant showed an increased number of inter-links with FUS_m in condensates, which suggests a stronger interaction between FUS_m and the HspB8-K141E mutant (*Figure 6E*, *Supplementary file 6*). A quantitative analysis revealed that almost all mono-links within the RRM domain of FUS_m were decreased upon binding of the HspB8-K141E mutant, suggesting that accessibility to these sites was hindered (*Figure 6F*, *Supplementary file 6*). Concomitantly, multiple intra-links within HspB8 bridging the N-terminus to the αCD domain were increased in the HspB8-K141E mutant (*Figure 6F*), an observation that is in line with a potential conformational change of the chaperone mutant upon FUS binding that brings these domains closer to each other (*Sailer et al., 2018*).

## Discussion

In this study, we show that the molecular chaperone HspB8 can prevent a disease-associated aberrant phase transition that is mediated by the protein FUS. We show that HspB8 uses its disordered domain to partition into liquid FUS condensates and that HspB8 uses a similar molecular grammar as described previously for FUS (*Wang et al., 2018*). More specifically, arginine residues in the IDR of HspB8 interact with tyrosine residues in the FUS-LCD, thereby promoting the targeting of the HspB8-αCD into FUS condensates for chaperoning of the misfolding-prone RRM domain of FUS.

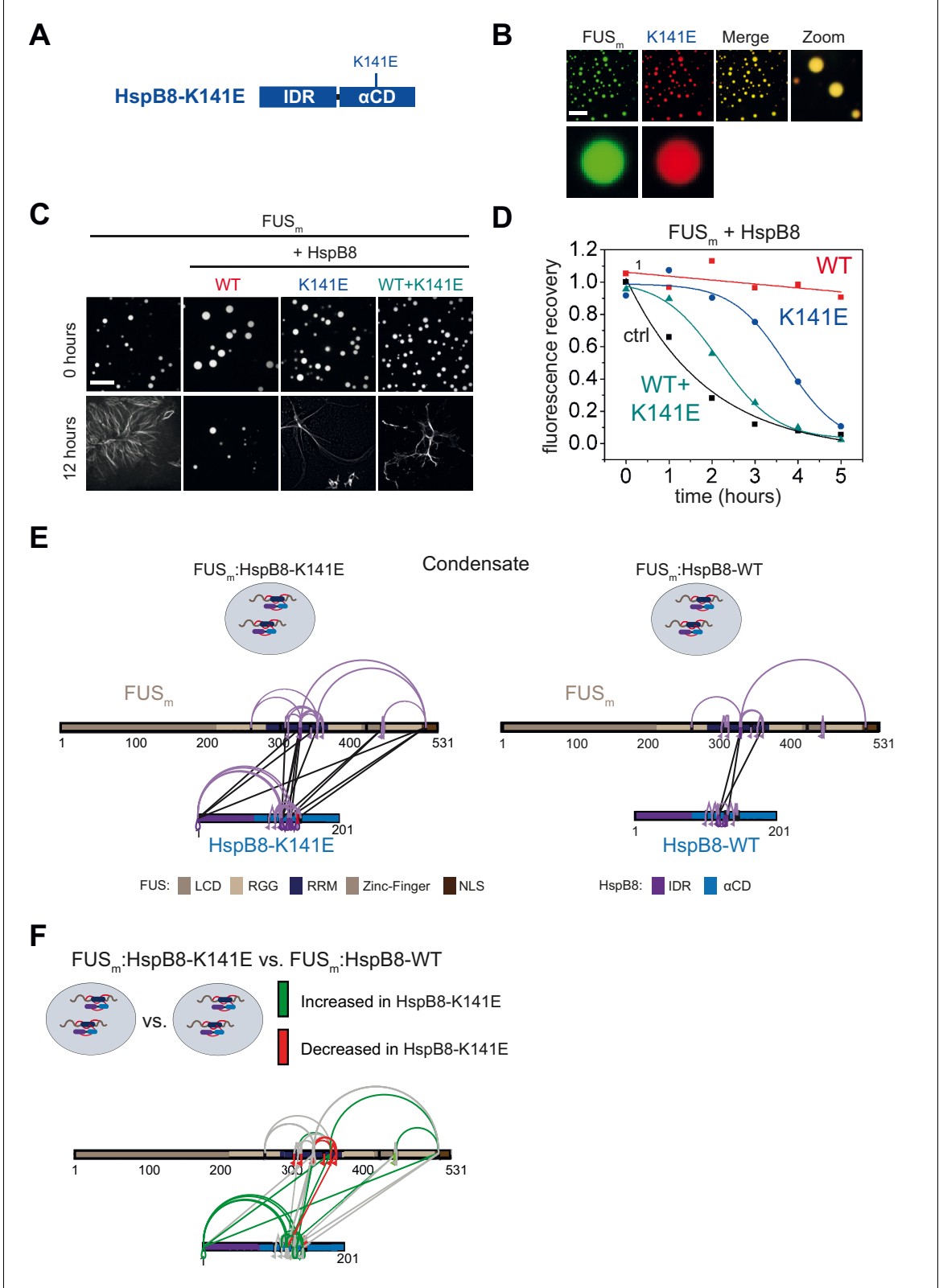

**Figure 6.** A disease-related mutation interferes with HspB8 activity. (**A**) Location of the disease-related K141E mutation inside the αCD of HspB8. (**B**) Partitioning of 5 µM HspB8-K141E (1:20 mix of labeled:unlabeled) into condensates formed by 5 µM FUS_m. Scale bar is 10 µm. (**C**) Aging assay of 5 µM FUS_m condensates in the absence (ctrl) and presence of HspB8-WT, HspB8-K141E, or a 1:1 mix of WT and K141 mutant. Final chaperone concentrations in all reactions are 125 nM. Scale bar is 10 µm. (**D**) Kinetics of the aging process. Plotted are the initial slopes of the FRAP recovery curves for FUS_m

*Figure 6 continued on next page*

*Figure 6 continued*

condensates in the absence (ctrl) and presence of HspB8-WT (red), HspB8-K141E (blue), or a 1:1 mix of WT and K141E mutant (cyan). (E) Equal amounts of FUS$_m$:HspB8-K141E (left) or FUS$_m$:HspB8-WT (right) were crosslinked under condensate inducing low salt conditions (75 mM NaCl). Experiments were carried out in triplicates and crosslinks were only considered, if they were identified in two out of three replicates with a deltaS<0.95, a minimum ld score$\geq$20, and an ld score$\geq$25 in at least one replicate (filtering was done on the level of the unique crosslinking site) and an FDR<0.05. The mutated site in HspB8-K141E is shown in red. Inter-links are shown in black and intra-links are shown in violet. Mono-links are shown with a flag, loop links with a pointed triangle, and homo-dimeric links with a loop. (F) Quantitative comparison of crosslinking patterns from HspB8-K141E and HspB8-WT condensates. Crosslinking sites that were upregulated or downregulated twofold or more (log2ratio$\geq$1 or $\leq-1$ and FDR$\leq$0.05) were considered significant and are highlighted in green (i.e., relative enrichment in FUS$_m$:HspB8-K141E condensates) or red (i.e., relative decrease in FUS$_m$:HspB8-K141E condensates). All other changes in crosslinking abundances were considered insignificant and are shown in gray background. $\alpha$CD, $\alpha$-crystallin domain; FRAP, fluorescence recovery after photobleaching.

This suggests a general principle for how the protein quality control machinery could be targeted to condensates in order to regulate misfolding-prone protein domains inside liquid condensates.

Despite the extensive work both in vivo and in vitro on proteins that phase separate, our current knowledge on how proteins are organized within condensates is limited and monitoring the transient interactions inside condensates has remained a major challenge to the field. In principle, proteomics and mass spectrometry should be an appropriate method to map condensate-specific interactions but reports using MS to study condensates remain scarce (*Xiang et al., 2015*; *Kadavath et al., 2015*; *Sanulli et al., 2019*). Recent studies demonstrate that proximity labeling in combination with MS is well suited to track the protein content of a specific molecular condensate (*Markmiller et al., 2018*; *Youn et al., 2018*), but falls short in charting direct PPIs or determining exact interaction sites. We and others showed previously that XL-MS is well suited to map PPIs and that relative changes in crosslinking as probed by quantitative XL-MS can provide a structural understanding of protein dynamics (*Yu and Huang, 2018*; *Sailer et al., 2018*; *Walker-Gray et al., 2017*; *Patel et al., 2017*; *Iacobucci et al., 2020*). Here, we adopt XL-MS to study condensates. In doing so we show with unprecedented molecular detail how protein contacts are formed within molecular condensates and demonstrate for the first-time condensate-specific client interaction.

Although the role of sHSPs in maintaining correct protein folding is intensively studied, up to now, the substrates of HspB8 have remained enigmatic (*Mymrikov et al., 2017*). Our crosslinking data suggests that HspB8 exerts its effect in part through the FUS-RRM domain. Misfolding of the RRM domain may represent the initial step on the pathway to forming a seed that subsequently promotes the nucleation of FUS fibrils. This may involve cross-beta sheet interactions of the LCDs via local concentration and LCD alignment. In this model, HspB8 would stabilize the fold of the RRM domain by binding to it, and by doing so maintain the liquid state of the condensates (*Figure 7*).

RRM domains are small folded entities that bind to single-stranded RNA. In humans, there are currently 745 RRM domains distributed over 446 proteins known to exist (*Letunic and Bork, 2018*). It has been proposed before that cells use RNA to regulate condensate formation (*Maharana et al., 2018*). Our data suggests that RNA and HspB8 can compete for their interaction with FUS. While this requires experimental validation in future studies, we hypothesize that some RRM domains require stabilization by binding to RNA, and in cases where no RNA is available, the chaperone HspB8 takes over this function and protects the RRM domain from unfolding and aggregation. Thus, it is a possibility that HspB8 has a general role in cell physiology in stabilizing RRM domains in condensates.

The changes we find in the inter-link patterns between FUS$_m$ and HspB8-WT or the neuropathy-causing mutant HspB8-K141E may appear counter-intuitive at first sight as far as they suggest a higher binding affinity of the mutant for FUS than the HspB8-WT. However, as sHSPs form oligomers subunit exchange between the WT and mutant HspB8 may influence the activity of the HspB8-WT. A heterodimer with lower activity than a WT homodimer could explain the differences in activity, and we do not exclude the possibility that HspB8 (which is present in mM concentrations inside the condensates, *Figure 2—figure supplement 1C and D*) will form higher-order oligomers which may enhance the mutant's effect on the WT. Our data also point to shifts in conformation and dynamics of the RRM domain after condensation and during aging and within the HspB8-K141E mutant upon RRM binding. These shifts may differentially affect the fate of the bound substrate: stabilization of the FUS$_m$ native conformation by HspB8-WT versus loss of function by the HspB8-K141E mutant

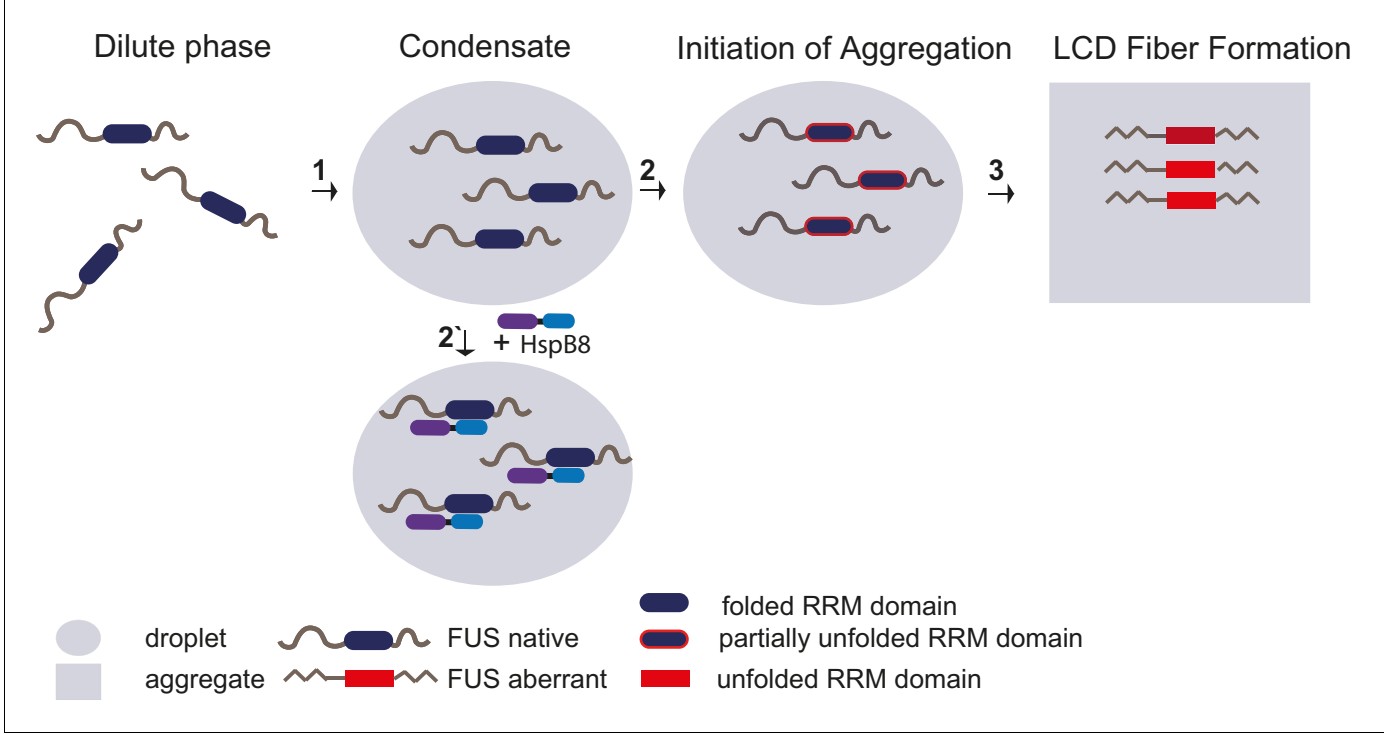

**Figure 7.** Unfolding of globular domains nucleates aberrant seeds for aggregation. Proposed mechanism where unfolding of globular domains in otherwise disordered proteins nucleates aberrant seeds for aggregation. In this model, the unfolding of the RRM domain comprises the initial step to seed the conversion of FUS into an aberrant conformational space that consequently involves the beta-amyloid formation of the LCDs by rising their local concentration and potentially by sterically aligning them for nucleation. Molecular chaperones like HspB8 stabilize the fold of the RRM domain and maintain the dynamic liquid state of the condensate. LCD, low complexity domain; RRM, RNA recognition motif.

with subsequent $FUS_m$ aggregation. To date, mutations in the genes coding for HspB1, HspB3, HspB5, and HspB8 have been associated with neuromuscular diseases (*Vendredy et al., 2020*). To what extent similar mechanisms may also affect other HspB8 mutants therefore awaits further investigation.

A mechanism by which unfolding of globular domains is controlled by chaperones inside condensates may be a process of general importance (*Franzmann and Alberti, 2019*). Deletion of the RRM slows down hardening but does not stop it. This suggests other driving forces such as aberrant LCD-LCD interactions that may require other types of chaperone systems to prevent them. Our study also highlights the interplay between folded domains and IDRs. A changed environment after phase separation could destabilize folded domains and require chaperones to stabilize them. Such a mechanism would necessitate many domain-specific chaperones for stabilization inside condensates.

In summary, by adapting existing XL-MS workflows, we were able to monitor PPIs and protein dynamics inside reconstituted protein condensates, thus paving the way for a deeper and more detailed structural understanding of condensate formation and aberrant phase transitions. Our study shows with unprecedented molecular detail how protein contacts are formed between a chaperone and the folded RRM of its client protein inside a condensate and therefore suggests a blueprint for how chaperones could act to stabilize biomolecular condensates in cells. More generally, our data suggests that established principles of cellular organization, such as domain-specific PPI sites, also apply to the biochemistry inside molecular condensates. Here, further work is required to expand the resolution of crosslinking mass spectrometry by developing crosslinking chemistries optimized for intrinsically disordered domains and to augment its current ability to also target condensates in vivo.

# Materials and methods

## Key resources table

| Reagent type (species) or resource | Designation | Source or reference | Identifiers | Additional information |
|---|---|---|---|---|
| Cell line (*Homo-sapiens*) | Hela Kyoto FUS-GFP BAC line | PMID:26496610 PMID:28377462 | 5063 | |
| Cell line (*Homo-sapiens*) | Hela Kyoto WT | PMID:26496610 | 5067 | |
| Cell line (*Escherichia coli*) | BL21-AI One Shot | Thermo Fisher Scientific | Cat# C607003 | Chemically competent *E. coli* |
| Chemical compound, drug | DSS-H12/D12 (BisSulfo Succinimidyl Suberate) | Creative Molecules Inc | Cat# 001S | |
| Chemical compound, drug | N,N-Dimethyl formamide (DMF) | Sigma-Aldrich | Cat# D4551 | |
| Chemical compound, drug | Ammonium bicarbonate | Sigma-Aldrich | Cat# 09830 | |
| Chemical compound, drug | Urea | Sigma-Aldrich | Cat# U5378 | |
| Chemical compound, drug | DTT (1,4-Dithiothreito) | Carl Roth | Cat# 6908.4 | |
| Chemical compound, drug | Trypsin | Promega | Cat# V5113 | |
| Chemical compound, drug | TCEP (Tris(2-carboxyethyl) phosphine hydrochloride) | Sigma-Aldrich | Cat# C4706-2G (CAS 51805-45-9) | |
| Chemical compound, drug | Iodoacetamide | Sigma-Aldrich | Cat# I1149-5G (CAS 144-48-9) | |
| Chemical compound, drug | Sep-Pak (C18) | Waters | Cat# WAT054960 | |
| Recombinant DNA reagent | FUS-GFP (plasmid) | PMID:26317470 | | FUS-GFP expression in Sf9 cells |
| Recombinant DNA reagent | FUS-G156E-GFP (FUS$_m$) (plasmid) | PMID:26317470 | | FUS-G156E-GFP expression in Sf9 cells |
| Recombinant DNA reagent | FUS-G156E-K9-GFP (FUS$_m$_K9) (plasmid) | This paper | | FUS-K9-GFP expression in Sf9 cells |
| Recombinant DNA reagent | FUS-G156E-ΔRRM-GFP (FUS$_m$ΔRRM) (plasmid) | This paper | | FUS-G156E-ΔRRM-GFP expression in Sf9 cells |
| Recombinant DNA reagent | FUS-17Y-GFP (plasmid) | This paper | | FUS-GFP-17Y expression in Sf9 cells |
| Recombinant DNA reagent | FUS-LCD-GFP (plasmid) | PMID:29961577 | | FUS-LCD-GFP expression in Sf9 cells |
| Recombinant DNA reagent | HspB8-WT (plasmid) | PMID:15879436 | | HspB8-WT expression in *E. coli* |
| Recombinant DNA reagent | HspB8-K141E (plasmid) | This paper | | HspB8-K141E expression in *E. coli* |
| Recombinant DNA reagent | HspB8-SNAP (FL) (plasmid) | This paper | | HspB8-SNAP expression in *E. coli* |

*Continued on next page*

*Continued*

| Reagent type (species) or resource | Designation | Source or reference | Identifiers | Additional information |
|---|---|---|---|---|
| Recombinant DNA reagent | HspB8-IDR-SNAP (IDR) (plasmid) | This paper | | HspB8-IDR-SNAP expression in *E. coli* |
| Recombinant DNA reagent | HspB8-αCD-SNAP (αCD) (plasmid) | This paper | | HspB8-αCD-SNAP expression in *E. coli* |
| Recombinant DNA reagent | SNAP (plasmid) | This paper | | SNAP expression in *E. coli* |
| Recombinant DNA reagent | HspB8-IDR-HspB1-αCD (IDR8αCD1) (plasmid) | This paper | | HspB8-IDR-HspB1-αCD expression in *E. coli* |
| Recombinant DNA reagent | HspB1-IDR-HspB8-αCD (IDR8αCD1) (plasmid) | This paper | | HspB1-IDR-HspB8-αCD expression in *E. coli* |
| Recombinant DNA reagent | HspB8-0R (plasmid) | This paper | | HspB8-0R expression in *E. coli* |
| Recombinant DNA reagent | HspB8-WT-3xmyc (plasmid) | This paper | | HspB8-WT-3xmyc expression in human cells |
| Recombinant DNA reagent | HspB8-0R-3xmyc (plasmid) | This paper | | HspB8-0R-3xmyc expression in human cells |
| Antibody | Anti-c-myc (monoclonal) | Santa Cruz Biotechnology | Clone 9E10 | IF(1:100) |
| Antibody | Anti-eIF4G (monoclonal) | Santa Cruz Biotechnology | Clone H-300 | IF(1:100) |
| Antibody | Anti-mouse Alexa Fluor 594 (monoclonal) | Thermo Fisher Scientific | A-21203 | IF(1:1000) |
| Antibody | Anti-rabbit Alexa Fluor 594 (monoclonal) | Thermo Fisher Scientific | A-21206 | IF(1:1000) |
| Software, algorithm | xQuest 2.1.3 | PMID:24356771 | http://proteomics.ethz.ch/cgi-bin/xquest2_cgi/download.cgi | |
| Software, algorithm | xProphet 2.1.3 | PMID:24356771 | http://proteomics.ethz.ch/cgi-bin/xquest2_cgi/download.cgi | |
| Software, algorithm | xTract 1.0.2 | PMID:26501516 | http://proteomics.ethz.ch/cgi-bin/xtract_cgi/index.cgi | |
| Software, algorithm | xiNET | PMID:25648531 | http://crosslinkviewer.org/ | |
| Software, algorithm | python 3.7.2 and 3.7.6 | | https://python.org | |
| Software, algorithm | pandas 1.0.3 | | https://pandas.pydata.org | |
| Software, algorithm | altair 4.1.0 | *Altair Developers, 2020* | https://altair-viz.github.io | |
| Software, algorithm | seaborn 0.9.0 | | https://seaborn.pydata.org | |
| Software, algorithm | Fiji | PMID:22743772 | https://imagej.net/software/fiji/ | |
| Other | Superdex Peptide 3.2/30 | GE Healthcare | Cat# 29-0362-31 | |
| Other | Acclaim PepMap RSLC | Thermo Fisher Scientific | Cat# P/N 164943 | |

*Continued on next page*

*Continued*

| Reagent type (species) or resource | Designation | Source or reference | Identifiers | Additional information |
|---|---|---|---|---|
| Other | EASY-nLC 1200 system | Thermo Fisher Scientific | LC140 | |
| Other | Orbitrap Fusion Tribrid Mass Spectrometer | Thermo Fisher Scientific | | |
| Other | Deposited Data: MS raw files | PRIDE https://www.ebi.ac.uk/pride/ | PXD021114 PXD021115 | |

## Protein expression and purification

FUS-G156E-GFP (FUS$_m$) was purified as described (*Patel et al., 2015*). HspB8 and corresponding variants were subcloned in a pET11d vector as N-terminal 3C protease-cleavable GST fusion proteins. Fusion proteins were expressed and purified from BL21 Codon RIL (Stratagene). Expression was induced by adding 0.15 mM IPTG for 4.5 hr at 37°C. Bacteria were lysed in 1× PBS, 5 mM DTT, 1 mM EDTA with EDTA-free Protease inhibitors tablet (Roche), and GST purified. Eluates were dialyzed with a 3500 Da MWCO membrane against 1× PBS, 5 mM DTT, and cleaved with PreScission protease. Cleaved off GST was removed by reverse GST purification. HspB8 proteins were subjected to ResourceQ ion-exchange chromatography, concentrated, dialyzed to HspB8 buffer (20 mM Tris, pH 7.4, 20 mM KCl, and 1 mM DTT), and validated by MS.

## In vitro experiments

Frozen aliquots of FUS$_m$ were thawed for 10 min at room temperature (RT), cleared from aggregates by centrifugation for 1 min at 21,000×$g$ using a 0.2 µm spin filter device. Molecular aging experiments were performed according to *Alberti et al., 2018* at 5 µM FUS$_m$ in reaction buffer (20 mM Tris-HCl, pH 7.4, 75 mM KCl, 0.75% Glycerol, and 1 mM DTT). FRAP experiments were performed and analyzed according to *Patel et al., 2015* at 5 µM FUS$_m$ in reaction buffer. For partitioning experiments, HspB8 and variants thereof were labeled with Cyanine-3-monosuccinimidyl ester (AAT bioquest, ABD-141) at equimolar ratio in HspB8 buffer and excess dye was removed by dialysis against HspB8 buffer with 1 mM DTT. Labeled HspB8 was mixed with unlabeled protein at a molar ratio of 1:20 and 5 µM FUS$_m$ was mixed with 5 µM total HspB8 in reaction buffer. Samples were applied into an imaging chamber with a coverslip passivated with polyethylene glycol. Fluorescence and DIC microscopy were performed on a confocal spinning disk microscope. Images were analyzed using Fiji software (*Schindelin et al., 2012*).

## Optical tweezer experiments

To characterize the material state of FUS$_m$ condensates with or without HspB8, controlled fusion experiments were performed in a custom-build dual-trap optical tweezer microscope (*Patel et al., 2015*; *Jahnel et al., 2011*). 5 µM FUS$_m$ condensates were phase-separated at T0 in reaction buffer with or without 20 µM HspB8 and immediately applied to a sample chamber. Two condensate droplets were trapped in two optical traps of the same trap stiffness at low overall light intensity to minimize local heating. With the first trap stationary, the second trap was moved to bring the droplets into contact and initiate coalescence, after which both traps were kept stationary. Laser signals and bright-field microscopy images were simultaneously recorded. Signals from the two traps—equal in magnitude, opposite in sign—were combined into the differential signal, from which coalescence relaxation times were deduced (*Wang et al., 2018*). To quantify the coalescence dynamics and account for droplets of different sizes, the relaxation time was normalized by the geometric radius of the two fusing droplets. Successful droplet coalescence was scored as yes (*Banani et al., 2017*) or no (0) depending on whether the process resulted in a near-spherical shape of the final droplet within 60 s. This duration was an order of magnitude longer than the earliest coalescence relaxation

times under all conditions. Coalescence success/failure data of $FUS_m$ without HspB8 were fit with a logistic regression model to estimate the half-life of liquid-like $FUS_m$ condensates.

## Time-lapse microscopy of fiber growth

$FUS_m$ was aged in a centrifuge tube for >24 hr to allow most of the protein to convert to fibrous/aged material. A small amount of aged material was flowed into a custom-built flow cell which includes upper and lower glass surfaces; the bottom glass surface was passivated with polyethylene glycol. After incubation, the chamber was flushed with freshly formed $FUS_m$ condensates in reaction buffer either in the presence or absence of 20 µM HspB8. An image stack representing a volume of approximately 100 $\mu m^3$ and a voxel size of 0.1 µm×0.1 µm×0.3 µm was acquired every 30 min using a spinning disk confocal equipped with a glycerol immersion 60× objective. In the resulting image stacks, fibers tend to be relatively dim with bright cores. To produce an image that allows for good visualization of the process of fiber growth, we smoothed each stack and subsequently applied an enhanced local contrast method (CLAHE). This method uses tiles throughout the image and calculates an appropriate contrast for each tile. We used CLAHE implemented in ImageJ with a block size of 30. For each stack, we create a maximum projection of the resulting stack that produces a single image. Finally, we used an image registration method (using the stackreg plugin in ImageJ) to remove any small translational drift which occurs through the process. The resulting movie is shown as *Video 1*.

## Image analysis to identify droplets and fibers

The identification of fibers and droplets in images was carried out using custom-made scripts in MATLAB. In short, each image is resized eight times using a bicubic interpolation. An image is subsequently automatically thresholded using Otsu's method through the imthresh command. Objects are identified as regions of connected pixels. Any objects which intersect the picture border or are very small are discarded from further analysis. Fibrous objects are identified as either objects with an eccentricity above 0.7 or, if the eccentricity is low, as objects that have a roughness above 6.4 pixels. The eccentricity is found by fitting an ellipse to a connected region and is defined as the ratio of the distance between the foci of the ellipse and its major axis length (implemented using the eccentricity argument in the regionprops command). To determine the roughness, each object is fit by a circle. The roughness is defined as the mean distance between the object border and the circular fit. Objects with low surface roughness as well as low eccentricity were considered droplets. All objects and their subsequent classification are also reviewed finally by eye to ensure that the parameters for the images are set properly.

## Immunostaining of sHSPs in stressed cells

HeLa Kyoto WT (Identifier 5067) and HeLa Kyoto FUS-GFP (Identifier 5063) BAC cells were cultured in Dulbecco's modified Eagle's medium containing 4.5 g/L glucose (Gibco Life Technologies) supplemented with 10% fetal bovine serum, 100 U/ml Penicillin+100 µg/ml Streptomycin. 250 µg/ml Geneticin (all Gibco Life Technologies) was added to the HeLa Kyoto FUS-GFP BAC cells. Cells were maintained at 37°C in a 5% $CO_2$ incubator (Thermo Fisher Scientific). HeLa Kyoto FUS-GFP BAC cells were described previously (*Mateju et al., 2017*). For immunostaining of HspB8-WT and HspB8-0R in stress granules, HeLa Kyoto cells were transfected with 200 ng of plasmids coding for HspB8-0R-3xmyc or HSPB8-WT-3xmyc using Lipofectamine 2000 (Life Technologies) following the manufacturer's instructions. 24 hr post-transfection, cells were subjected to heat shock in a water bath at 43.5°C for 1 hr. Cells were fixed with 3.7% formaldehyde for 9 min at RT and permeabilized with acetone for 5 min at −20°C and stained with c-Myc (9E10, Santa Cruz Biotechnology) and eIF4G (H-300, Santa Cruz Biotechnology) specific antibodies. Secondary antibodies used were anti-mouse Alexa Fluor 594 (A-21203, Thermo Fisher Scientific) and anti-rabbit Alexa Fluor 488 (A-21206, Thermo Fisher Scientific).

## Crosslinking of molecular condensates

Frozen aliquots of $FUS_m$ and $FUS_{m\_}K9$ protein stored in 50 mM Tris-HCl pH 7.5, 500 mM KCl, 5% Glycerol, and 1 mM DTT were thawed for 10 min at RT, cleared from aggregates by centrifugation for 1 min at 21,000×$g$ using a 0.2 µm spin filter device and subsequently diluted in water to a low

salt solution (final concentration of 75 mM KCl) to induce phase separation or into a high salt solution (final concentration of 500 mM KCl) to prevent phase separation. In order to reconstitute FUS$_m$: HspB8 condensates HspB8-WT or HspB8-K141E mutant were added to FUS$_m$ condensates at equal mass ratio and subsequently incubated on ice for 10 min to allow for sufficient mixing. Molecular condensates were crosslinked by addition of 0.9 mM H12/D12 DSS (Creative Molecules) (*Figure 2E*, *Figure 2—figure supplement 1I*), 1.9 mM (*Figures 1B, C*, *6E and F*, *Figure 2—figure supplement 1F and G*), or 1.5 mM (*Figure 5D*) at a molar ratio crosslinker to lysines of ~3.6, ~2.3, or ~3.6, respectively, for 30 min at 37°C shaking at 650 rpm in a Thermomixer (Eppendorf). Protein samples were quenched by the addition of ammonium bicarbonate to a final concentration of 50 mM and either directly evaporated to dryness or after an additional centrifugation step for 60 min at 21,000×*g* in order to separate the dense phase of the condensates from the dilute phase. The dilute phase containing supernatant was transferred to a fresh tube and both phases were evaporated to dryness. Thus, samples were subjected to MS analysis either directly after adjustment of the high salt/low salt solution containing mixtures of essentially dilute phase or dense phase of the condensates (*Figures 1B, C*, *5B*, *6E and F*, *Figure 2—figure supplement 1F & G*, *Figure 3—figure supplement 1G*) or after an additional separating step by centrifugation fully separating reconstituted droplets that had formed under low salt solution from the remaining dilute phase as described above (*Figures 2E*, *5B and D*, *Figure 2—figure supplement 1I*, *Figure 3—figure supplement 1G*). For a detailed description, also of the lysine-rich variant FUS_K9 see also legend *Figure 2—figure supplement 1G*.

## Crosslinking coupled to mass spectrometry (XL-MS)

Crosslinked samples were processed essentially as described (*Leitner et al., 2014*). In short, the dried protein samples were denatured in 8 M Urea, reduced by the addition of 2.5 mM TCEP at 37°C for 30 min, and subsequently alkylated using 5 mM Iodacetamid at RT for 30 min in the dark. Samples were digested by the addition of 2% (w/w) trypsin (Promega) overnight at 37°C after adding 50 mM ammonium hydrogen carbonate to a final concentration of 1 M urea. Digested peptides were separated from the solution and retained by a C18 solid-phase extraction system (SepPak Vac 1cc tC18 [50 mg cartridges, Waters]) and eluted in 50% ACN, 0.1% FA. After desalting the peptides were evaporated to dryness and stored at −20°C. Dried peptides were reconstituted in 30% ACN, 0.1% TFA, and then separated by size exclusion chromatography on a Superdex 30 increase 3.2/300 (GE Life Science) to enrich for crosslinked peptides. The three early-eluting fractions were collected for MS measurement, evaporated to dryness, and reconstituted in 5% ACN, 0.1% FA. Concentrations were normalized by A215 nm measurement to ensure equal amounts of dilute and dense phase and peptides separated on a PepMap C18 2 µM, 50 µM×150 mm (Thermo Fisher Scientific) using a gradient of 5–35% ACN for 45 min. MS measurement was performed on an Orbitrap Fusion Tribrid mass spectrometer (Thermo Fisher Scientific) in data dependent acquisition mode with a cycle time of 3 s. The full scan was done in the Orbitrap with a resolution of 120,000, a scan range of 400–1500 m/z, AGC Target 2.0e5, and injection time of 50 ms. Monoisotopic precursor selection and dynamic exclusion were used for precursor selection. Only precursor charge states of 3–8 were selected for fragmentation by collision-induced dissociation using 35% activation energy. MS2 was carried out in the ion trap in normal scan range mode, AGC target 1.0e4, and injection time of 35 ms. Data were searched using *xQuest* in ion-tag mode. Carbamidomethylation (+57.021 Da) was used as a static modification for cysteine. As database the sequences of the measured recombinant proteins along with reversed and shuffled sequences were used for the FDR calculation by *xProphet*.

Experiments were carried out in three biologically independent sets of experiments (meaning separate batches of expressed protein). For one set of experiments, each sample was independently crosslinked in triplicates and each of these was measured in technical duplicates. Crosslinks were only considered, if they were identified in two out of three replicates with a deltaS<0.95, a minimum Id score≥20, and an ld score≥25 in at least one replicate (filtering was done on the level of the unique crosslinking site), and an FDR≤0.05 as calculated by *xProphet* for at least one replicate.

## Quantitation of crosslinked peptides from condensates (qXL-MS)

### Quantitation

Initial processing of identified crosslinked peptides for quantitation was performed essentially as described (*Sailer et al., 2018*). In short, the chromatographic peaks of identified crosslinks were integrated and summed up over different peak groups for quantification by *xTract* (taking different charge states and different unique crosslinked peptides for each unique crosslinking site into account). Only high-confidence crosslinks that fulfilled the above introduced criteria were selected for further quantitative analysis.

The resulting *bagcontainer.details.stats.xls* file was used as an input for in-house scripts developed for this manuscript. The bag container contains all experimental observations on a peptide level as extracted by *xTract* (e.g., peptide mass, charge state, the extracted MS1 peak area, and any violations assigned by *xTract*). Missing observations were replaced by imputation with random values drawn from a normal distribution based on our experimental distribution. Here, the log-normal experimental distribution of measured MS1 peak areas was converted to a normal distribution by log2-conversion. Of the resulting normal distribution, the mean and standard deviations were determined. The mean was shifted downward while the width was decreased in order to obtain the distribution to draw imputed values from, following the same procedure and parameters as described for Perseus (*Tyanova et al., 2016*) (width: 0.3 and down shift: 1.8).

Data were additionally filtered using a light-heavy filter as described (*Walzthoeni et al., 2015*) and peptides with a light-heavy log2ratio$<-1$ or $>1$ were excluded from further analysis. Experiments were normalized by their mean MS1 peak area using the mean of all experiments as reference. The ratio of each experiment compared to the reference was computed and all observed MS1 areas were multiplied by this experiment-specific ratio to receive the same mean for all experiments. In addition, replicates were normalized within each experiment. Thus, the mean of each biological and technical replicate within an experiment was shifted to the mean of an experiment in the same way as described above.

In the next step, log2ratios were calculated as the difference between the log2-converted MS1 peak areas (instead of the ratio). Here, the MS1 area for each experiment was shifted into a log2 scale after all summing operations but before taking any means, allowing us to calculate meaningful standard deviations between biological replicates and to avoid the influence of outliers in the original log-normal scale. P-value calculations were otherwise performed as described (*Walzthoeni et al., 2015*), with one notable exception: MS1 peak areas were not split by technical replicates in order to avoid artificially improved p-values with increasing numbers of technical replicates. FDR values were p-values corrected for multiple testing, following the Benjamini–Hochberg procedure.

### Significance

Only high-confidence crosslinking sites (see above) that were detected reliably and consistently with a deltaS$<0.95$, a minimum Id score$\geq20$, and a Id score$\geq25$ in at least one replicate (filtering was done on the level of the unique crosslinking site), and an FDR$\leq0.05$ were used for quantitation. Changes in crosslinking abundances were throughout the paper quantified against the dilute phase (i.e., relative enrichment within droplets is shown in green; relative decrease in red). Only crosslinking sites that were upregulated or downregulated twofold or more (log2ratio$\geq1$ or $\leq-1$ and FDR$\leq0.05$) in at least two biological replicate sets of experiments and in addition contained no opposing regulation in any replicate set were considered significant.

## Time-resolved quantitative crosslinking coupled to mass spectrometry

Fresh FUS$_m$ condensates formed under low salt (75 mM KCl) conditions were left shaking at 650 rpm in a Thermomixer (Eppendorf) at 28°C and monitored by fluorescence microscopy at regular intervals until conversion into fibers. The stock solution was aliquoted prior to dilution into low salt buffer to induce condensation and aliquots (n=3) were crosslinked for 5 min and flash-frozen in liquid nitrogen at indicated time points: T1–T6 (0 hr, 20 min, 40 min, 1 hr, 1 hr 20 min, 1 hr 40 min; condensates), T7–T9 (2 hr 20 min, 2 hr 40 min, 3 hr; gels), and T10–T11 (12 hr and 24 hr; fibers). While thawing, 1M ammonium bicarbonate was added to a final concentration of 50 mM and samples were evaporated to dryness. Crosslinks were subsequently identified and quantified exactly as described above.

## Visualization of Crosslink data

In order to both validate and visualize the crosslink information, multiple in-house scripts have been written. The visualization scripts either interface directly with the quantitation script described above or use *xTract*-like output for an input. In either case, the filtering and significance criteria as described above for the quantitation script were used. Crosslink data were transformed via pandas (version 1.0.3) for assessment. *Figures 1B* and *5B*, *Figure 3—figure supplement 1G* were created with altair (version 4.1.0) running on Python version 3.7.6. *Figure 1C* and *Figure 1—figure supplement 1A* (left and right Panel) were created with seaborn (version 0.9.0) running on python (version 3.7.2). *Figure 1—figure supplement 1A* (middle panel) was created using the in-built pandas dataframe plot functions.

## RNA competition assay

FUS$_m$:HspB8 condensates were prepared as described and either incubated with RNA oligonucleotide PrD (*Maharana et al., 2018*) in sub-stoichiometric amounts (three times molar excess of FUS$_m$) or equal volume of water. Samples were checked by microscopy before crosslinking to ensure that the addition of RNA did not dissolve the condensates. The dense phase of the condensates was separated from the dilute phase by centrifugation and the concentration normalized prior to MS measurement as described above.

## Acknowledgements

The authors thank the Protein Expression Purification and Characterization Facility, the Light Microscopy Facility and the Mass Spectrometry Facility at the Max Planck Institute of Molecular Cell Biology and Genetics in Dresden, the Bioimaging Centre (BIC) at the University of Konstanz, and the Centro Interdipartimentale Grandi Strumenti (CIGS) at the University of Modena and Reggio Emilia. The authors thank Marit Leuschner and Anne Schwager for their assistance with cell culture. The authors thank members of the Alberti, Hyman and Stengel Labs, and Dewpoint Therapeutics for helpful discussions and critical reading of the manuscript. This work was supported by the Konstanz Research School Chemical Biology (KoRS-CB) and the Cluster of Excellence 'Physics of Life', TU Dresden, Dresden, Germany. SA and SC are grateful for the EU Joint Programme—Neurodegenerative Disease Research (JPND) project. SC acknowledges funding from AriSLA Foundation (Granulopathy and MLOpathy), MAECI (Dissolve_ALS), and MIUR (Departments of excellence 2018–2022; E91I18001480001). FS is funded by the Emmy Noether Programme of the DFG (STE 2517/1-1).

## Additional information

### Competing interests

Edgar E Boczek, Ina Poser: is currently an employee of Dewpoint Therapeutics. Simon Alberti: is a shareholder, consultant and member of the scientific advisory board for Dewpoint Therapeutics. Anthony A Hyman: is cofounder, shareholder, consultant and member of the scientific advisory board for Dewpoint Therapeutics. Florian Stengel: is a consultant and member of the scientific advisory board for Dewpoint Therapeutics. The other authors declare that no competing interests exist.

### Funding

| Funder | Grant reference number | Author |
|---|---|---|
| Deutsche Forschungsgemeinschaft | STE 2517/1-1 | Florian Stengel |
| Universität Konstanz | Chemicals and small Equipment Purchase | Florian Stengel |
| Deutsche Forschungsgemeinschaft | Cluster of Excellence "Physics of Life" | Simon Alberti Anthony A Hyman |
| EU Joint Programme – Neurodegenerative Disease Research | Neurodegenerative Disease Research (JPND) | Serena Carra Simon Alberti |

| Fondazione Italiana di Ricerca per la Sclerosi Laterale Amio-trofica | Granulopathy and MLOpathy | Serena Carra |
| Ministero degli Affari Esteri e della Cooperazione Internazio-nale | Dissolve_ALS | Serena Carra |
| Ministry of Education, University and Research | E91I18001480001 | Serena Carra |

The funders had no role in study design, data collection and interpretation, or the decision to submit the work for publication.

## Author contributions

Edgar E Boczek, Julius Fürsch, Conceptualization, Formal analysis, Investigation, Methodology, Writing - original draft, Writing - review and editing; Marie Laura Niedermeier, Conceptualization, Formal analysis, Validation, Investigation, Visualization, Methodology, Writing - review and editing; Louise Jawerth, Formal analysis, Validation, Investigation, Methodology, Writing - review and editing; Marcus Jahnel, Martine Ruer-Gruß, Jie Wang, Xiao Yan, Andrej Pozniakovski, Ina Poser, Daniel Mateju, Lars Hubatsch, Investigation; Kai-Michael Kammer, Software, Formal analysis, Investigation, Visualization; Peter Heid, Software, Formal analysis, Validation, Investigation, Visualization; Laura Mediani, Validation, Investigation; Serena Carra, Conceptualization, Supervision, Funding acquisition; Simon Alberti, Anthony A Hyman, Florian Stengel, Conceptualization, Formal analysis, Supervision, Funding acquisition, Validation, Investigation, Visualization, Methodology, Writing - original draft, Project administration, Writing - review and editing

## Author ORCIDs

Lars Hubatsch https://orcid.org/0000-0003-1934-7437
Florian Stengel https://orcid.org/0000-0003-1447-4509

## Decision letter and Author response

Decision letter https://doi.org/10.7554/eLife.69377.sa1
Author response https://doi.org/10.7554/eLife.69377.sa2

# Additional files

## Supplementary files

• Supplementary file 1. Crosslink identification and quantification, related to *Figure 1A and B* and *Figure 1—figure supplement 1A*.

• Supplementary file 2. Crosslink identification and quantification for FUS aging experiment, related to *Figure 1C* and *Figure 1—figure supplement 1C, D, E and F* and *Figure 3—figure supplement 1H and I*.

• Supplementary file 3. Crosslink identification, related to *Figures 2E* and *5B* and *Figure 2—figure supplement 1E, F, G and I* and *Figure 3—figure supplement 1G*.

• Supplementary file 4. Crosslink quantification, related to *Figure 5B* and *Figure 3—figure supplement 1G*.

• Supplementary file 5. Crosslink quantification, related to *Figure 5D* and *Figure 5—figure supplement 1C*.

• Supplementary file 6. Crosslink identification and quantification, related to *Figure 6E and F*.

• Transparent reporting form

## Data availability

All data generated or analysed during this study are included in this published article (and its supplementary information files). The MS data (raw files, xQuest, xTract and in-house quantitation output

files) have been deposited to the ProteomeXchange Consortium via the PRIDE partner repository with the dataset identifier PXD021114 and PXD021115.

The following datasets were generated:

| Author(s) | Year | Dataset title | Dataset URL | Database and Identifier |
|-----------|------|---------------|-------------|-------------------------|
| Boczek EE | 2021 | HspB8 prevents aberrant phase transitions of FUS by chaperoning its folded RNA binding domain | https://www.ebi.ac.uk/pride/archive/projects/PXD021114 | PRIDE, PXD021114 |
| Boczek EE | 2021 | HspB8 prevents aberrant phase transitions of FUS by chaperoning its folded RNA binding domain | https://www.ebi.ac.uk/pride/archive/projects/PXD021115 | PRIDE, PXD021115 |

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
