## [Decision Letter]

**Acceptance summary:**

The transition from a more liquid like phase to a solid phase of liquid-liquid phase separated condensates is an important disease mechanism. Here the authors use an innovative approach to study the effect of chaperones to counteract this aging process which is of great importance for understanding the molecular details of aggregation prone diseases. The methodological novelty of this approach is the use of protein-protein crosslinking to study the conformation of the RNA-binding domain of FUS in condensates in the absence and presence of HspB8.

**Decision letter after peer review:**

Thank you for submitting your article "HspB8 prevents aberrant phase transitions of FUS by chaperoning its folded RNA binding domain" for consideration by *eLife*. Your article has been reviewed by 3 peer reviewers, and the evaluation has been overseen by Volker Dötsch as the Senior and Reviewing Editor and Reviewer #3. The following individuals involved in review of your submission have agreed to reveal their identity: Henning Urlaub (Reviewer #1); Justin L.P. Benesch (Reviewer #2).

Essential revisions:

1) How is it possible to discriminate between intramolecular and intermolecular crosslinks, especially when the crosslinks were created within aggregates of proteins such as, in this case, FUS? Does the RRM indeed change its conformation, or does it simply interact with RRMs of other FUS molecules? Is the spacer of DSS not too long to allow any conclusion about the structure of such small domains as the RRM or Zn-finger, in particular for a protein prone to aggregation?

2) The authors titrated the crosslinker according to the numbers of lysine residues present in the two proteins HspB8 and FUS. They should discuss, or at least state, the concentration of the crosslinking reagent within the condensates compared with the concentration outside the condensate phase; this statement should include the correlation between the concentration of the crosslinker in the condensate and the protein concentration(s) inside and outside the condensates. (The authors used a 2.7 fold molar excess of crosslinker over lysine residues in the proteins – however, they should also state the concentrations as molarity.) A very high concentration of a chemical crosslinker could in principle grossly distort the structure of a protein, as has been observed in several studies; this might diminish the value of crosslinking as an indicator of protein condensation, and the authors should address this. Monitoring crosslinking in a concentration-dependent manner would have also been beneficial.

3) The exclusive use of a lysine-reactive crosslinker limits the resolution of monitoring intra- and inter-protein interactions. Although the authors state explicitly that they use a lysine-directed crosslinker solely because there are no lysine residues in the disordered regions of FUS, it is not clear why the authors did not also use crosslinkers reactive toward groups other than the amino group. Such crosslinkers could possibly have been useful to study protein interactions that are salt-sensitive. Indeed, the LCD of FUS is not suitable for applying lysine-reactive or carboxy-reactive crosslinkers; better suited to this would be (for example) ruthenium (II) tris-bipyridine, which connects tyrosine residues, and it is precisely tyrosine(s) in the LCD that seem(s) to be crucial for protein condensation. Furthermore, protein-protein crosslinking by UV light has been described and is attributed to covalent Y-Y links.

In particular: (i) Have the authors considered the use of different endoproteinases (such as chymotrypsin) in order to obtain a comprehensive sequence coverage of crosslinked peptides? (ii) Why have the authors not taken into account serine, threonine and tyrosine as crosslinking sites of DSS. It has been reported that DSS is also reactive toward these amino acids.

Related: The LCD of FUS contains the N-terminus of the protein – a primary amine. Are no cross-links seen to this?

4) Figure 5A: droplets formed by FUSm-ΔRRM are significantly larger than those formed by FUSm. Addition of Hsp8B seem to make them larger still (clearly obvious for the "36h" time point). Some questions are raised: What fraction of the FUSm is present the droplets/fibres and what fraction remains in solution? Is HspB8 present in the condensates as well? Complementary experiments using labelled HspB8 are necessary to clarify this.

5) The hypothesis that RNA and HspB8 compete for the interaction with FUS is interesting; however, the data supporting this are preliminary and the question requires more thorough investigation for example by pulldown experiments.

6) Can the authors rule out that the differences seen between FUS in low and high salt is not due to the salt concentration per se, but due to the phase separation? While unlikely, but it could be that DSS does not penetrate the FUS-rich phase, and cross-links are only observed between species in the FUS-poor phase (which still contains FUS, albeit at lower concentrations). Can the authors rule this out?

7) The ACD-snap and IDR-snap experiments are a bit confusing – the ACD doesn't partition, yet appears to have some activity? While the IDR region doesn't partition nearly as well as the full-length, the N-terminal regions of sHSPs are frequently unstable c.f. aggregation – are the authors sure that the IDR-snap construct is stable – and if not, might this explain the difference?

8) HSPB8 likes to dimerize. Nonetheless, it is hard to explain the "dominant negative" effect seen in the mixture of WT/mutant. Do the authors have an explanation – perhaps due to some higher-order oligomerisation..? Or difference in partitioning..?

Related: While most of the experiments described in tis manuscript provide detailed structural, mechanistic information, the paragraph on the CMT mutant HspB8-K141E is less insightful and remains pretty vague. It seems counterintuitive that stronger binding of the mutant prevents the aging of the droplets and the transition to a more solid-like phase. Here more data or at least a more detailed discussion with a reasonable model would be helpful.

9) The authors analysed the data using the search engine xQuest, which requires a pair of non- and isotopically labelled crosslinkers for confident identification of crosslinked peptides. The data already acquired are probably unsuitable for other crosslink search engines (35 ms ion accumulation time and ion-trap analyser for MS2). In general, I consider the crosslinking data acquired in this work by CID in the ion trap to be of insufficient quality. Have the authors attempted to acquire a data set with high-resolution MS2 and to use more modern search engines with relaxed crosslinker specificity?

---

## [Author Response]

Essential revisions:1) How is it possible to discriminate between intramolecular and intermolecular crosslinks, especially when the crosslinks were created within aggregates of proteins such as, in this case, FUS? Does the RRM indeed change its conformation, or does it simply interact with RRMs of other FUS molecules? Is the spacer of DSS not too long to allow any conclusion about the structure of such small domains as the RRM or Zn-finger, in particular for a protein prone to aggregation?

This is an important remark by the reviewer. We cannot generally discriminate between intra crosslinks and interlinks between homo-oligomers and we also did not want to be suggestive of this; we can however discriminate for a specific subset of those.

We had tried to get this important point across in the legend to Figure S1A:

“XL-MS cannot readily discriminate if a crosslink has formed within one polypeptide chain (defined as intra-link, vide supra) or between homodimers or even higher oligomers of the same protein. This is usually not a problem, but in the case of the high protein concentrations within condensates it may play a role. However, as we detect virtually all shown intra-links within FUS_m_ and HspB8 also under the relatively low-concentration regime of the dilute phase, it is fair to assume that at least a major part of the observed changes occurs within intra-links as usually defined – e.g. occurring within one FUS_m_ polypeptide chain – as we assume in the current version of the manuscript. For some specific intra-links, we do however know that they must have occurred between different molecules of the same protein; these are crosslinks between overlapping peptides whose sequence is unique within the protein and that must therefore originate from different copies of the same protein (homodimeric link).”

These dimeric links are also indicated by a “d” in all figures – see for example Figure 1B.

At least this subset of homodimeric links – some of which we can follow in detail during both condensation and aging (see Figure 1B and 1C) – tells us that there must be heightened contacts between different RRM molecules within the dense phase of the condensates (which was admittedly also not an entirely unexpected result, given the high concentration of the condensates). Moreover, we also detect virtually all links specified as intralinks already in the dilute phase, suggesting that their increase can at least not only be due to heightened RRM contacts between different FUS molecules in the dense phase of the condensates. The same line of arguments can be applied to the aging time-course, as we observe here a decrease in dimeric intra-links and have also detected virtually all intra-links both in the dilute and dense phase (see Figure S1D), which again indicates that they should at least partially have emanated from monomeric intra-links.

We would also like to point out that we are very aware of the fact of that molecular condensates are particularly critical protein assemblies that require extra care. This is why we have analyzed critical experiments in high redundancy, e.g. as biological triplicate *experiments*, that each consist of 3 independently crosslinked replicates (which were additionally all measured as technical duplicates) (see Figure 2E).

We also fully agree with the reviewer that it is not possible to directly assign a specific conformational status to a crosslinking pattern, even though our work has demonstrated that changes in intralink pattern *can* reflect conformational changes (PMID: 30361475). This is why we tried very carefully throughout our manuscript to NOT directly assign a specific conformation status with the observed change in crosslinking pattern within the RRM.

For example:

“This suggests that there are augmented contacts within the RRM domain, indicative of a structural change. In addition, there was an increase in homo-dimeric links between RRM domains, indicative of interactions between RRM domains of different FUS_m_ molecules.”

However, our main point is, that the majority of changes in crosslink abundance that we observe takes place WITHIN the RRM.

This fact, in combination with all the follow-up biochemical and molecular characterization on the folding status of the RRM in Figure 5 and 6, lead us to conclude that there is likely indeed a conformation change taking place in the RRM during condensation and in particularly during aging – which can be prevented by binding of HspB8; which is again in line with our crosslinking data, as we for example show in Figure S3I that the crosslinking pattern within the RRM does NOT change after binding of HspB8.

We also respectfully disagree that the linker that was employed in this study is too long to allow any conclusion about the structure of such small domains as the RRM or Zn-finger. We know from previous work that the linker used in this study covers distances up to 30 Å well, with a particular focus on distances from 12 to 15 Å (PMID: 22286754).

These are also the likely distances to be covered within the RRM and between HspB8 and FUS- RRM and FUS ZnF, as for example mapping our identified crosslinks to the existing NMR structure of the RRM of FUS (PDB 2LCW) demonstrates an excellent fit (i.e. we can map 8 out of 8 identified crosslinks, covering distances from 12 to 31 Å) and shows that also small distances can be covered by our crosslinker DSS H_12_/D_12_ (Author response figure 1).

**Author response image 1. sa2fig1:** Mapping identified cross-links identified using the crosslinker DSS H_12_/D_12_ onto the NMR structure of the RRM of FUS (PDB 2LCW).

2) The authors titrated the crosslinker according to the numbers of lysine residues present in the two proteins HspB8 and FUS. They should discuss, or at least state, the concentration of the crosslinking reagent within the condensates compared with the concentration outside the condensate phase; this statement should include the correlation between the concentration of the crosslinker in the condensate and the protein concentration(s) inside and outside the condensates. (The authors used a 2.7 fold molar excess of crosslinker over lysine residues in the proteins – however, they should also state the concentrations as molarity.) A very high concentration of a chemical crosslinker could in principle grossly distort the structure of a protein, as has been observed in several studies; this might diminish the value of crosslinking as an indicator of protein condensation, and the authors should address this. Monitoring crosslinking in a concentration-dependent manner would have also been beneficial.

We apologize for this oversight and have now specified and corrected the used molar excess of added crosslinker for each experiment and have also added molar concentrations (see updated experimental procedures). It is unfortunately not possible to directly measure the concentration of the crosslinker within the dense phase of the condensate and the dilute phase. We are nevertheless confident that this has not been a problem, as our experiments demonstrate that we obtain a similar amount of intralinks within the dense phase of the condensate and the dilute phase (see Figure 2E, Figures S2 F, G and E), which is a good indication that sufficient amounts of crosslinker were present in either of the phases.

We also agree with the reviewer that there are reports showing that an excess of crosslinker can lead to distortions in protein conformation. However, we respectfully disagree that high concentrations of a chemical crosslinker regularly or even generally distort the structure of a protein. More importantly, we did not employ high doses of crosslinker in our study, as we were operating within a regime of ~ 1 to 2 mM crosslinker (see updated experimental procedures for details) and such a conventional and widely accepted crosslinker concentration range (PMID:31045356). Moreover, as the protein concentration within the dense phase of the condensates is even higher (Figures S2C and S2D), the likelihood of possible distortions is even lower there.

As suggested by the reviewer, we have now monitored crosslinking in a concentration-dependent manner and our data shows that there are no apparent differences in crosslink behavior and crosslinking yields within the crosslinker concentration range that was used in this study – i.e. 1 to 2 mM DSS (Author response image 2).

**Author response image 2. sa2fig2:** Monitoring crosslinking in a concentration-dependent shows no apparent differences in crosslink behavior and crosslinking yield in the range used in this study, i. e. 1 to 2 mM DSS (see updated experimental procedures).

3) The exclusive use of a lysine-reactive crosslinker limits the resolution of monitoring intra- and inter-protein interactions. Although the authors state explicitly that they use a lysine-directed crosslinker solely because there are no lysine residues in the disordered regions of FUS, it is not clear why the authors did not also use crosslinkers reactive toward groups other than the amino group. Such crosslinkers could possibly have been useful to study protein interactions that are salt-sensitive. Indeed, the LCD of FUS is not suitable for applying lysine-reactive or carboxy-reactive crosslinkers; better suited to this would be (for example) ruthenium (II) tris-bipyridine, which connects tyrosine residues, and it is precisely tyrosine(s) in the LCD that seem(s) to be crucial for protein condensation. Furthermore, protein-protein crosslinking by UV light has been described and is attributed to covalent Y-Y links.In particular: (i) Have the authors considered the use of different endoproteinases (such as chymotrypsin) in order to obtain a comprehensive sequence coverage of crosslinked peptides? (ii) Why have the authors not taken into account serine, threonine and tyrosine as crosslinking sites of DSS. It has been reported that DSS is also reactive toward these amino acids.Related: The LCD of FUS contains the N-terminus of the protein – a primary amine. Are no cross-links seen to this?

We had taken known side-reactions to serine, tyrosine and threonine into account – without any significant gain in information in regard to covering the LCD (Author response image 3).

**Author response image 3. sa2fig3:** Comparison of crosslinks within FUS when exclusively lysine-lysine crosslinks (upper panel) or additionally side reactions to serine, tyrosine and threonine are taken into account (lower panel) (n=3).

Taking side-reactions into account did indeed increase the overall number of detected cross-links, including links within the IDR of HspB8 and the ZnF of FUS; however, the overall gain in information was not significant and comes, at least in our experience, at the prize of a higher likelihood of false-assigned crosslinks. We had therefore decided to not consider side-reactions for our study (Supplementary Data1 Revision).

We had also carried out UV-linking using Sulfo-SDA in combination with chymotrypsin as alternative endoproteinase with no significant gain in information, as it worked very well for HspB8, but unfortunately not for FUS and thus decided to not employ it in our study (Author response image 4). Still, and reassuringly, also in this case our finding that crosslinks formed between the αCD of HspB8 and the RRM of FUS_m_ within condensates could be confirmed.

**Author response image 4. sa2fig4:** Overall crosslinking pattern from a mixture of recombinant FUS_m_ and HspB8 under condensate inducing low salt (75 mM KCl) conditions that were UV crosslinked using Sulfo-SDA and searched using *StavroX/MeroX* (including side reactions to serine, tyrosine and threonine) after digestion with chymotrypsin.

We had also used chymotrypsin and AspN as additional endoproteinases in our “standard” approach using DSS and the *xQuest/xProphet* pipeline. While the use of chymotrypsin worked well for HspB8, we did unfortunately detect only a rather limited number of crosslinks in FUS, which is why we again decided to not use it for our study (Author response image 5). Similar results had been achieved for AspN (Author response image 5). Still, we were again able to detect crosslinks that had formed between the αCD of HspB8 and the RRM of FUS_m_.

**Author response image 5. sa2fig5:** Overall crosslinking pattern from mixtures of recombinant FUSm and HspB8 under condensate inducing low salt (75 mM NaCl) conditions that were digested using either chymotrypsin (interlinks are shown in black and Intralinks, monolinks and loop links in violet (n=3)) or AspN as endoproteinases (all links are shown in orange (n=1)).

This lack of suitable endoproteinases for in-depth characterization of the FUS-LCD unfortunately also limited the use of covalent Y-Y linkers, as suggested by the reviewer:

The reviewer is also fully correct in pointing out that the LCD of FUS contains a primary amine in its N-terminus and we did indeed find crosslinks to it. We nevertheless had decided to leave this data out as we have no valid information on the LCD otherwise – see above.

For all these reasons we had designed the FUS_m__K9 mutant, as it allowed for a full coverage of the LCD. Excitingly, it confirmed our finding that the crosslinks that had formed between the αCD of HspB8 and the RRM of FUSm were *condensate-specific* (Figure S2G).

Details of the FUS K9_mutant and the obtained results can be found in the respective figure legend Figure S2G:

“As FUS_m_ contains no lysine residues within its LCD which are amenable to our NHS-ester crosslinker, we generated a FUS_m_ variant with an additional “26” lysines in the LCD (FUS_m__K9), which resembles FUS_m_ in its condensation behavior even though it does not exhibit a similar ageing phenotype. We find that in the dense phase of the condensates the number of inter-links between the LCD of FUS_m_ and both the IDR and the αCD of HspB8 was strongly increased, confirming previous results on the role of the LCD during phase separation. However, only inter-links between the αCD of HspB8 and the RRM of FUS_m_ were exclusively detected inside the condensates.”

4) Figure 5A: droplets formed by FUSm-ΔRRM are significantly larger than those formed by FUSm. Addition of Hsp8B seem to make them larger still (clearly obvious for the "36h" time point). Some questions are raised: What fraction of the FUSm is present the droplets/fibres and what fraction remains in solution? Is HspB8 present in the condensates as well? Complementary experiments using labelled HspB8 are necessary to clarify this.

We have now compared the propensity to form droplets of the FUSm-ΔRRM mutant with FUS_m_ (Author response image 6), demonstrating that FUSm-ΔRRM has a lower csat than FUS_m_ and explaining why droplets formed by FUSm-ΔRRM are significantly larger than those formed by FUS_m_.

**Author response image 6. sa2fig6:** Comparison of condensation behavior of FUS_m_ and FUSm-ΔRRM.

Addition of Hsp8B which is present in mM concentrations inside the condensates (Figures S2C and S2D) adds a lot of material to the droplets, which explains their enlargement.

As suggested by the reviewer we have also carried out complementary experiments using labelled HspB8, which we have now added as Figure S5A in the manuscript, which clearly show that HspB8 was present in the condensates as well.

5) The hypothesis that RNA and HspB8 compete for the interaction with FUS is interesting; however, the data supporting this are preliminary and the question requires more thorough investigation for example by pulldown experiments.

We agree with the reviewer that the current data in support of our hypothesis that RNA and HspB8 compete for their interaction with FUS is somewhat preliminary and requires experimental validation in future studies. However, interaction/binding studies as suggested by the reviewer are very challenging in condensates and will likely not be quantitative. For example, we have tried to monitor both HspB8 levels in the dilute and dense phase by SDS-PAGE after adding different amounts of RNA and also tried to monitor aging of FUS_m_ and FUS_m_∆RRM under increasing amounts of RNA. However, as the amount of RNA itself influences condensate formation and as condensate formation is additionally somewhat dependent on the exact composition of the pool of added RNAs, the data was inconclusive and did not permit a quantitative statement. We therefore feel that although the suggested experiment is interesting in its own right, it is clearly beyond the scope of the present manuscript.

However, in order to get this important point across more clearly (i.e. the current rather hypothetical status of our investigations and the need for future investigations), we have amended the manuscript to write:

“This result suggests that RNA and HspB8 may compete for binding to FUS_m_ and indicate a similar mechanism by which they stabilize its RRM domain to prevent FUS_m_ aging.”

and extended our discussion to write:

“It has been proposed before that cells use RNA to regulate condensate formation. Our data suggests that RNA and HspB8 can compete for their interaction with FUS. While this requires experimental validation in future studies, we hypothesize that some RRM domains require stabilization by binding to RNA and in cases where no RNA is available, the chaperone HspB8 takes over this function and protects the RRM domain from unfolding and aggregation.”

We hope that this is a satisfactory solution for the reviewer; however, in case the reviewer should insist, we would be willing to completely remove Figure 5D from our manuscript prior to publication.

6) Can the authors rule out that the differences seen between FUS in low and high salt is not due to the salt concentration per se, but due to the phase separation? While unlikely, but it could be that DSS does not penetrate the FUS-rich phase, and cross-links are only observed between species in the FUS-poor phase (which still contains FUS, albeit at lower concentrations). Can the authors rule this out?

In our workflow we have included a centrifugation step *after* we crosslink to fully separate the dense phase of the condensates from the dilute phase. This extra step guarantees that we analyze dilute phase and dense separately and that the crosslinks detected in the dense phase must indeed stem from proteins that were located within the condensate (Figure 2E).

In these experiments we did also NOT change the salt concentration (i.e. we only used the low salt concentration of 75 mM to induce condensation in the first place). This already rules out that the differences in cross-links were due to a change in salt concentration. Moreover, in our control experiments (see Figure S2F) – where we indeed used low salt vs high-salt conditions without additional centrifugation – we identify the same crosslinking pattern (including the *condensate-specific* enrichment of links between the HspB8-αCD and the FUS_m_-RRM) as we detect without centrifugation, again corroborating that our crosslinker is able to penetrate the FUS-rich phase and that differences in crosslinking are not due to a change in salt concentration. The comparability of our “centrifugation“ and „salt“ experiments is additionally underpinned by Figure S3G which shows that also our quantitative crosslinking data are highly comparable between these conditions.

We apologize that this important point was not evident from the manuscript and was likely caused by an inadequate description of our experimental procedures, which we have therefore amended.

7) The ACD-snap and IDR-snap experiments are a bit confusing – the ACD doesn't partition, yet appears to have some activity? While the IDR region doesn't partition nearly as well as the full-length, the N-terminal regions of sHSPs are frequently unstable c.f. aggregation – are the authors sure that the IDR-snap construct is stable – and if not, might this explain the difference?

The residual activity of the ACD-SNAP construct may be explained by the fact that this variant somewhat partitions into the FUS droplets as shown in Figure S4E and into FUS-LCD droplets as shown in Figure S4H. ACD-SNAP still shows about 18% of partitioning in comparison to the full-length HspB8-SNAP (Figure S4E). Considering that the concentration of the full-length HspB8 protein inside the FUS condensates is about 2.4 mM (Figures S2C and S2D), a lowered partitioning of the ACD-SNAP variant to 18% reflects a concentration of 0.4 mM inside the condensate. This fact could well explain the observation that the ACD-SNAP shows residual chaperone activity towards FUS inside the condensates (Figure S4F). These results are further supported by the fact that the swap variant with a HspB1-IDR fused to the HspB8-ACD (IDR1ACD8) shows a similar effect (Figure S4M and S4N). We did not observe aggregation of the IDR-SNAP variant during protein purification or our experiments. A possible explanation of the results may be related to the oligomeric state of HspB8 inside the droplets. Dimerization/oligomerization of HspB8 may play a role in increasing multivalency of interactions with itself and with FUS and thus have an influence on its partition coefficient. As the IDR-SNAP construct lacks the dimerization competent αCD, this may contribute to a reduced partitioning of this variant.

8) HSPB8 likes to dimerize. Nonetheless, it is hard to explain the "dominant negative" effect seen in the mixture of WT/mutant. Do the authors have an explanation – perhaps due to some higher-order oligomerisation..? Or difference in partitioning..?Related: While most of the experiments described in tis manuscript provide detailed structural, mechanistic information, the paragraph on the CMT mutant HspB8-K141E is less insightful and remains pretty vague. It seems counterintuitive that stronger binding of the mutant prevents the aging of the droplets and the transition to a more solid-like phase. Here more data or at least a more detailed discussion with a reasonable model would be helpful.

It is indeed a well-known fact that sHSPs form oligomers and it is therefore a possibility that subunit exchange between the wildtype and mutant HspB8 takes place, which may influence the activity of the HspB8-WT. A heterodimer with lower activity than a wildtype homodimer could explain the differences in activity, and we do not exclude the possibility that HspB8 (which is present in mM concentrations inside the condensates, Figures S2C and S2D) will form higher order oligomers which may enhance the mutant’s effect on the wildtype.

Moreover, our data indicates that the mutant has a higher affinity for FUS than the wildtype (as suggested by the stronger binding) which could lead to an at least partial displacement of the HspB8 wildtype by the mutant. We reasoned that this in turn could result in a conformational change – or the stabilization of such a change – within FUS that affects its aging behavior. A notion that is again at least partially supported by changes that we detect in our crosslinking data that support a conformational change within the RRM of FUS during both condensation and aging (Figure 1B and 1C) and within the HspB8 mutant upon binding of the FUS RRM (Figure 6F).

“Concomitantly, multiple intra-links within HspB8 bridging the N-terminus to the αCD domain were increased in the HspB8-K141E mutant (Figure 6F), an observation that is in line with a potential conformational change of the chaperone mutant upon FUS binding that brings these domains closer to each other.”

The reviewer is however fully right in pointing out that currently this is mostly speculation and future research will be needed to shed light on this question. This is also why we write in the discussion:

“To what extent similar mechanisms may also affect other HspB8 mutants therefore awaits further investigation.”

In order to emphasize these important points even further, we have now removed any speculative parts from the Results section, and have amended the discussion of the manuscript to now write:

“The changes we find in the inter-link patterns between FUSm and HspB8-WT or the neuropathy-causing mutant HspB8-K141E may appear counter-intuitive at first sight as far as they suggest a higher binding affinity of the mutant for FUS than the HspB8-WT. However, as sHSPs form oligomers subunit exchange between the wildtype and mutant HspB8 may influence the activity of the HspB8-WT. A heterodimer with lower activity than a wildtype homodimer could explain the differences in activity, and we do not exclude the possibility that HspB8 (which is present in mM concentrations inside the condensates, Figures S2C and S2D) will form higher order oligomers which may enhance the mutant’s effect on the wildtype. Our data also point to shifts in conformation and dynamics of the RRM domain after condensation and during aging and within the HspB8-K141E mutant upon RRM binding. These shifts may differentially affect the fate of the bound substrate: stabilization of the FUSm native conformation by HspB8-WT versus loss of function by the HspB8-K141E mutant with subsequent FUSm aggregation. (….) To what extent similar mechanisms may also affect other HspB8 mutants therefore awaits further investigation.”

9) The authors analysed the data using the search engine xQuest, which requires a pair of non- and isotopically labelled crosslinkers for confident identification of crosslinked peptides. The data already acquired are probably unsuitable for other crosslink search engines (35 ms ion accumulation time and ion-trap analyser for MS2). In general, I consider the crosslinking data acquired in this work by CID in the ion trap to be of insufficient quality. Have the authors attempted to acquire a data set with high-resolution MS2 and to use more modern search engines with relaxed crosslinker specificity?

We respectfully disagree that crosslinking data acquired by CID in an ion trap leads per se to data of insufficient quality. A notion that is also shared in the latest and first community-wide crosslinking study (PMID:31045356), where the approach chosen in this manuscript is explicitly mentioned as being *lege artis*, *see* page 6958:

“Some strategies provide additional layers of evidence that can be used to better control the error rate. For example, isotope-coded, noncleavable linkers provide two independent measures of precursor and fragment masses and charge state information for fragments independent of MS resolution”.

We also would like to point out that during the time-point where we have started this project, the approach used by us was the only one available that allowed for the automatic quantification of crosslink data.

As suggested by the reviewer we have nevertheless acquired new data from samples of FUS::HspB8 condensates using high-resolution MS2 and searched them independently using the search engines *pLink2* (PMID: 31363125) and *MeroX* (PMID: 31283178) (Author response image 7). Author response image 7 shows an excellent overlap between the different approaches, where virtually all crosslinks detected by our approach could be validated by the high-resolution acquisition strategy and the different search engines. While *pLink2* detects overall more crosslinks than *xQuest*, the question if this is due to an improved acquisition strategy or rather a higher false-discovery rate, as shown previously for *pLink2* (PMID: 32029734), can remain open at this point, as the data clearly confirms that the data generated by the approach used for this manuscript consists of valid and highly reproducible crosslinks. A similar argument can be made for the comparison with *MeroX*, in which case *xQuest* is able to detect a larger number of crosslinks.

**Author response image 7. sa2fig7:** Comparison of detected crosslinks from FUS/HspB8 condensates at an FDR < 0. 05 using either a high/low acquisition strategy and *xQuest* as a search engine or a high/high strategy and *plink2* (A) or *MeroX* (B) as search engine. Please note that due to technical reasons the comparison with *MeroX* included only inter- and intraprotein crosslinks, while the comparison with *pLink2* additionally comprised mono links and loop links.

Moreover, to hopefully dissipate any remaining reservations of the reviewer we have additionally validated our data by parallel reaction monitoring (PRM). As the reviewer knows, this targeted approach is highly sensitive and will with very high certainty confirm if a specific peptide was present within a sample or not. We randomly chose 20 unique crosslinking sites (*uxIDs*) from our dataset and were able to *validate all the tested uxIDs from our dataset by PRM* (Author response image 8).

**Author response image 8. sa2fig8:** Shown are select transitions for crosslinked peptide pairs of FUS. In total, our assay contained 100 transitions and while a few isolated transitions did not result in conclusive traces, we could validate all tested *uxIDs* from our dataset by PRM.

We hope that this data convinces the reviewer not only that the crosslinking data acquired in this work is of sufficient quality but that acquiring crosslinking data using the ion trap and isotope-coded noncleanable linkers is generally a good strategy to obtain high-quality crosslink information from in-vitro condensates in order to probe protein-protein interactions.